# ONE-STEP DIFFUSION POLICY: FAST VISUOMOTOR POLICIES VIA DIFFUSION DISTILLATION

## ABSTRACT

Diffusion models, praised for their success in generative tasks, are increasingly being applied to robotics, demonstrating exceptional performance in behavior cloning. However, their slow generation process stemming from iterative denoising steps poses a challenge for real-time applications in resource-constrained robotics setups and dynamically changing environments. In this paper, we introduce the One-Step Diffusion Policy (OneDP), a novel approach that distills knowledge from pre-trained diffusion policies into a single-step action generator, significantly accelerating response times for robotic control tasks. We ensure the distilled generator closely aligns with the original policy distribution by minimizing the Kullback-Leibler (KL) divergence along the diffusion chain, requiring only 2%-10% additional pre-training cost for convergence. We evaluated OneDP on 6 challenging simulation tasks as well as 4 self-designed real-world tasks using the Franka robot. The results demonstrate that OneDP not only achieves state-of-the-art success rates but also delivers an order-of-magnitude improvement in inference speed, boosting action prediction frequency from 1.5 Hz to 62 Hz, establishing its potential for dynamic and computationally constrained robotic applications. A video demo is provided *here*, and the code will be publicly available soon.

## 1 INTRODUCTION

Diffusion models (Sohl-Dickstein et al., 2015; Ho et al., 2020) have emerged as a leading approach to generative AI, achieving remarkable success in diverse applications such as text-to-image generation (Saharia et al., 2022; Ramesh et al., 2022; Rombach et al., 2022), video generation (Ho et al., 2022; OpenAI, 2024), and online/offline reinforcement learning (RL) (Wang et al., 2022; Chen et al., 2023b; Hansen-Estruch et al., 2023; Psenka et al., 2023). Recently, Chi et al. (2023); Team et al. (2024); Reuss et al. (2023); Ze et al. (2024); Ke et al. (2024); Prasad et al. (2024) demonstrated impressive results of diffusion models in imitation learning for robot control. In particular, Chi et al. (2023) introduces the diffusion policy and achieves a state-of-the-art imitation learning performance on a variety of robotics simulation and real-world tasks.

However, because of the necessity of traversing the reverse diffusion chain, the slow generation process of diffusion models presents significant limitations for their application in robotic tasks. This process involves multiple iterations to pass through the same denoising network, potentially thousands of times (Song et al., 2020a; Wang et al., 2023). Such a long inference time restricts the practicality of using the diffusion policy (Chi et al., 2023), which by default runs at 1.49 Hz, in scenarios where quick response and low computational demands are essential. While classical tasks like block stacking or part assembly may accommodate slower inference rates, more dynamic activities involving human interference or changing environments require quicker control responses (Prasad et al., 2024). In this paper, we aim to significantly reduce inference time through diffusion distillation and achieve responsive robot control.

Considerable research has focused on streamlining the reverse diffusion process for image generation, aiming to complete the task in fewer steps. A prominent approach interprets diffusion models using stochastic differential equations (SDE) or ordinary differential equations (ODE) and employs advanced numerical solvers for SDE/ODE to speed up the process (Song et al., 2020a; Liu et al., 2022; Karras et al., 2022; Lu et al., 2022). Another avenue explores distilling diffusion models into

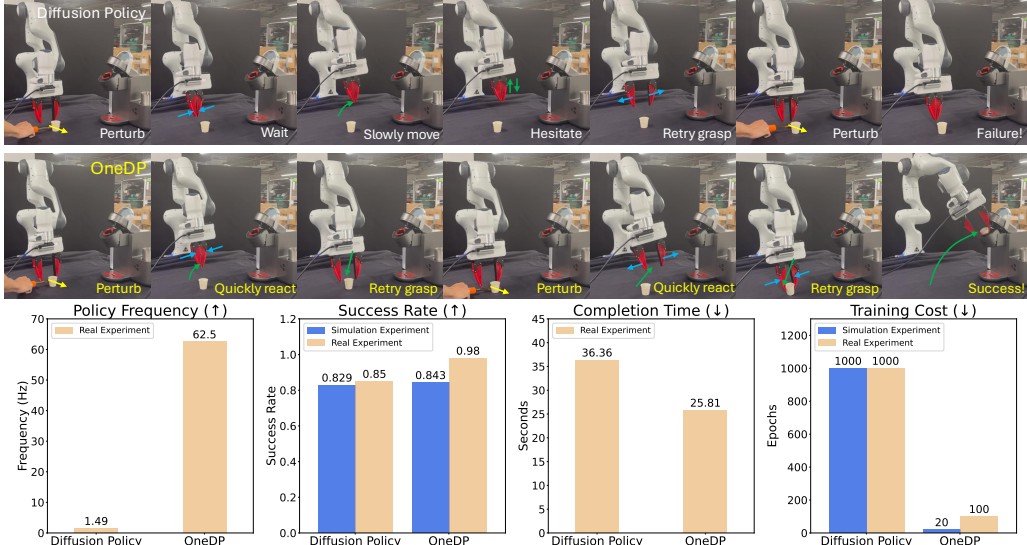

Figure 1: **Comparison of Diffusion Policy and One-Step Diffusion Policy (OneDP).** We demonstrate the rapid response of OneDP to changes in dynamic environments through real-world experiments. The first row illustrates how Diffusion Policy (Chi et al., 2023) struggles to adapt to environment changes (here, object perturbation) and fails to complete the task due to its slow inference speed. In contrast, the second row highlights OneDP's quick and effective response. The third row offers a quantitative comparison: in the first panel, OneDP executes action prediction much faster than Diffusion Policy. This enhanced responsiveness results in a higher average success rate across multiple tasks, particularly in real-world scenarios, as depicted in the second panel. The third panel reveals that OneDP also completes tasks more swiftly. The final panel indicates that distillation of OneDP requires only a small fraction of the pre-training cost.

generators that require only one or a few steps through Kullback-Leibler (KL) optimization or adversarial training (Salimans & Ho, 2022; Song et al., 2023; Luo et al., 2024; Yin et al., 2024). However, accelerating diffusion policies for robotic control has been largely underexplored. Consistency Policy (Prasad et al., 2024) (CP) employs the consistency trajectory model (CTM) (Kim et al., 2023a) to adapt the pre-trained diffusion policy into a few-step CTM action generator. Despite this, several iterations for sampling are still required to maintain good empirical performance.

In this paper, we introduce the One-Step Diffusion Policy (OneDP), which distills knowledge from pre-trained diffusion policies into a one-step diffusion-based action generator, thus maximizing inference efficiency through a single neural network feedforward operation. We demonstrate superior results over baselines in Figure 1. Inspired by the success of SDS (Poole et al., 2022) and VSD (Wang et al., 2024) in text-to-3D generation, we propose a policy-matching distillation method for robotic control. The training of OneDP consists of three key components: a one-step action generator, a generator score network, and a pre-trained diffusion-policy score network. To align the generator distribution with the pre-trained policy distribution, we minimize the KL divergence over diffused actions produced by the generator, with the gradient of the KL expressed as a score difference loss. By initializing the action generator and the generator score network with the identical pre-trained model, our method not only preserves or enhances the performance of the original model, but also requires only 2%-10% additional pre-training cost for the distillation to converge. We compare our method with CP and demonstrate that it outperforms CP with a higher success rate across tasks, leveraging a single-step action generator and achieving $20\times$ faster convergence. A detailed comparison with this approach is provided in Sections 3 and 4.

We evaluate our method in both simulated and real-world environments. In simulated experiments, we test OneDP on the six most challenging tasks of the Robomimic benchmark (Mandlekar et al., 2021). For real-world experiments, we design four tasks with increasing difficulty and deploy OneDP on a Franka robot arm. In both settings, OneDP demonstrated state-of-the-art success rates with single-step generation, performing $42\times$ faster in inference.

## 2 ONE-STEP DIFFUSION POLICY

### 2.1 PRELIMINARIES

Diffusion models are powerful generative models applied across various domains (Ho et al., 2020; Sohl-Dickstein et al., 2015; Song et al., 2020b). They function by defining a forward diffusion process that gradually corrupts the data distribution into a known noise distribution. Given a data distribution $p(\boldsymbol{x})$, the forward process adds Gaussian noise to samples, $\boldsymbol{x}^0 \sim p(\boldsymbol{x})$, with each step defined as $\boldsymbol{x}^k = \alpha_k \boldsymbol{x}^0 + \sigma_k \boldsymbol{\epsilon}_k$, where $\boldsymbol{\epsilon}_k \sim \mathcal{N}(\mathbf{0}, \boldsymbol{I})$. The parameters $\alpha_k$ and $\sigma_k$ are manually designed and vary according to different noise scheduling strategies.

A probabilistic model $p_\theta(\boldsymbol{x}^{k-1}|\boldsymbol{x}^k)$ is then trained to reverse this diffusion process, enabling data generation from pure noise. DDPM (Ho et al., 2020) uses discrete-time scheduling with a noise-prediction model $\epsilon_\theta$ to parameterize $p_\theta$, while EDM (Karras et al., 2022) employs continuous-time diffusion with $\boldsymbol{x}^0$-prediction. We use epsilon prediction $\epsilon_\theta$ in our derivation. The diffusion model is trained using the denoising score matching loss (Ho et al., 2020; Song et al., 2020b).

Once trained, we can estimate the unknown score $s(\boldsymbol{x}^k)$ at a diffused sample $\boldsymbol{x}^k$ as:

$$s(\boldsymbol{x}^k) = -\frac{\epsilon^*(\boldsymbol{x}^k, k)}{\sigma_k} \approx -\frac{\epsilon_\theta(\boldsymbol{x}^k, k)}{\sigma_k}, \tag{1}$$

where $\epsilon^*(\boldsymbol{x}^k, k)$ is the true noise added at time $k$ and we denote $s_\theta(\boldsymbol{x}^k) = -\frac{\epsilon_\theta(\boldsymbol{x}^k, k)}{\sigma_k}$. With a score estimate, clean data $\boldsymbol{x}^0$ can be sampled by reversing the diffusion chain (Song et al., 2020b). This requires multiple iterations through the estimated score network, making it inherently slow.

Wang et al. (2022); Chi et al. (2023) extend diffusion models as expressive and powerful policies for offline RL and robotics. In robotics, a set of past observation images, $\mathbf{O}$, is used as input to the policy. An action chunk, $\mathbf{A}$, which consists of a sequence of consecutive actions, forms the output of the policy. Diffusion policy is represented as a conditional diffusion-based action prediction model,

$$\pi_\theta(\mathbf{A}^0|\mathbf{O}) := \int \cdots \int \mathcal{N}(\mathbf{A}^K; \mathbf{0}, \boldsymbol{I}) \prod_{k=K}^{k=1} p_\theta(\mathbf{A}^{k-1}|\mathbf{A}^k, \mathbf{O}) d\mathbf{A}^K \cdots d\mathbf{A}^1, \tag{2}$$

The explicit form of $\pi_\theta(\mathbf{A}^0|\mathbf{O})$ is often impractical due to the complexity of integrating actions from $\mathbf{A}^K$ to $\mathbf{A}^1$. However, we can obtain action chunk samples from it by iterative denoising. More details are provided in Appendix D

### 2.2 ONE-STEP DIFFUSION POLICY

Action sampling through the vanilla diffusion policies is notoriously slow due to the need of tens to hundreds of iterative inference steps. The latency issue is critical for computationally sensitive robotic tasks or tasks that require high control frequency. Although employing advanced ODE solvers (Song et al., 2020a; Karras et al., 2022) could help speed up the sampling procedure, empirically at least ten iterative steps are required to ensure reasonable performance. Here, we introduce a training-based diffusion policy distillation method, which distills the knowledge of a pre-trained diffusion policy into a single-step action generator, enabling fast action sampling.

We propose a one-step implicit action generator $G_\theta$, from which actions can be easily obtained as follows,

$$\boldsymbol{z} \sim \mathcal{N}(\mathbf{0}, \boldsymbol{I}), \mathbf{A}_\theta = G_\theta(\boldsymbol{z}, \mathbf{O}). \tag{3}$$

We define the action distribution generated by $G_\theta$ as $p_{G_\theta}$. Assuming the existence of a pre-trained diffusion policy $\pi_\phi(\mathbf{A}|\mathbf{O})$ defined by Equation (2) and parameterized by $\epsilon_\phi$, its corresponding action distribution is denoted as $p_{\pi_\phi}$. Drawing inspiration from the success of SDS (Poole et al., 2022) and VSD (Wang et al., 2024) in text-to-3D applications, we propose using the following reverse KL divergence to align the distributions $p_{G_\theta}$ and $p_{\pi_\phi}$,

$$\mathcal{D}_{KL}(p_{G_\theta}||p_{\pi_\phi}) = \mathbb{E}_{\boldsymbol{z} \sim \mathcal{N}(\mathbf{0}, \boldsymbol{I}), \mathbf{A}_\theta = G_\theta(\boldsymbol{z}, \mathbf{O})} \left[ \log p_{G_\theta}(\mathbf{A}_\theta|\mathbf{O}) - \log p_{\pi_\phi}(\mathbf{A}_\theta|\mathbf{O}) \right].$$

It is generally intractable to estimate this loss by directly computing the probability densities, since $p_{G_\theta}$ is an implicit distribution and $p_{\pi_\phi}$ involves integrals that are impractical (Equation (2)). However,

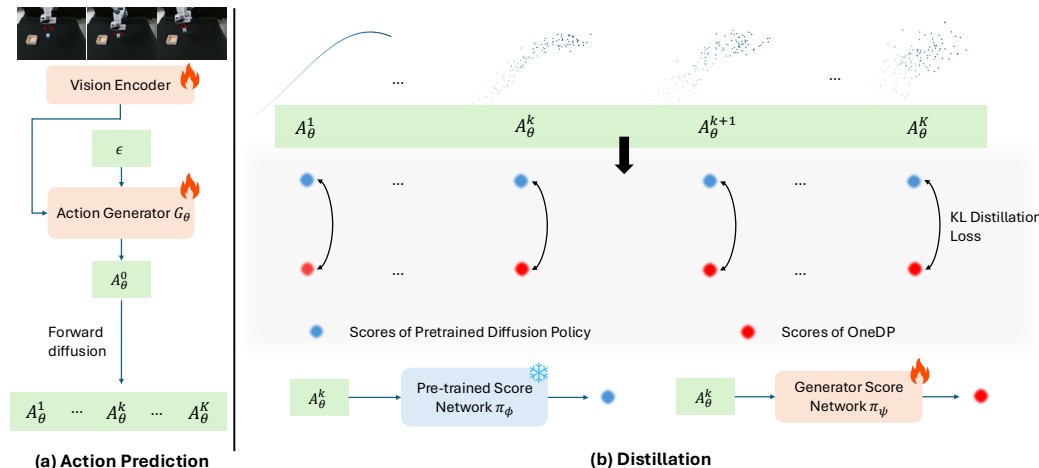

Figure 2: **Diffusion Distillation Pipeline.** a) Our one-step action generator processes image-based visual observations alongside a random noise input to deliver single-step action predictions. b) We implement KL-based distillation across the entire forward diffusion chain. Direct computation of the KL divergence is often impractical; however, we can effectively utilize the gradient of the KL, formulated into a score-difference loss. The pre-trained score network $\pi_\phi$ remains fixed while the action generator $G_\theta$ and the generator score network $\pi_\psi$ are trained.

we only need the gradient with respect to $\theta$ to train our generator by gradient descent:

$$\nabla_\theta \mathcal{D}_{KL}(p_{G_\theta}||p_{\pi_\phi}) = \mathbb{E}_{\substack{z \sim \mathcal{N}(\mathbf{0}, \mathbf{I}), \\ \mathbf{A}_\theta = G_\theta(z, \mathbf{O})}} \left[ (\nabla_{\mathbf{A}_\theta} \log p_{G_\theta}(\mathbf{A}_\theta|\mathbf{O}) - \nabla_{\mathbf{A}_\theta} \log p_{\pi_\phi}(\mathbf{A}_\theta|\mathbf{O})) \nabla_\theta \mathbf{A}_\theta \right]. \quad (4)$$

Here $s_{p_{G_\theta}}(\mathbf{A}_\theta) = \nabla_{\mathbf{A}_\theta} \log p_{G_\theta}(\mathbf{A}_\theta|\mathbf{O})$ and $s_{p_{\pi_\phi}}(\mathbf{A}_\theta) = \nabla_{\mathbf{A}_\theta} \log p_{\pi_\phi}(\mathbf{A}_\theta|\mathbf{O})$ are the scores of the $p_{G_\theta}$ and $p_{\pi_\phi}$ respectively. Computing this gradient still presents two significant challenges: First, the scores tend to diverge for samples from $p_{G_\theta}$ that have a low probability in $p_{\pi_\phi}$, especially when $p_{\pi_\phi}$ may approach zero. Second, the primary tool for estimating these scores, the diffusion models, only provides scores for the diffused distribution.

Inspired by Diffusion-GAN (Wang et al., 2023), which proposed to optimize statistical divergence, such as the Jensen–Shannon divergence (JSD), throughout diffused data samples, we propose to similarly optimize the KL divergence outlined in Equation (4) across diffused action samples as described below:

$$\nabla_\theta \mathbb{E}_{k \sim \mathcal{U}}[\mathcal{D}_{KL}(p_{G_\theta,k}||p_{\pi_\phi,k})] = \mathbb{E}_{\substack{z \sim \mathcal{N}(\mathbf{0}, \mathbf{I}), k \sim \mathcal{U} \\ \mathbf{A}_\theta = G_\theta(z, \mathbf{O}) \\ \mathbf{A}_\theta^k \sim q(\mathbf{A}_\theta^k|\mathbf{A}_\theta, k)}} \left[ w(k)(s_{p_{G_\theta}}(\mathbf{A}_\theta^k) - s_{p_{\pi_\phi}}(\mathbf{A}_\theta^k)) \nabla_\theta \mathbf{A}_\theta^k \right]. \quad (5)$$

where $w(k)$ is a reweighting function, $q$ is the forward diffusion process and $s_{p_{\pi_\phi}}(\mathbf{A}_\theta^k)$ could be obtained through Equation (1) with $\epsilon_\phi$. In order to estimate the score of the generator distribution, $s_{p_{G_\theta}}$, we introduce an auxiliary diffusion network $\pi_\psi(\mathbf{A}|\mathbf{O})$, parameterized by $\epsilon_\psi$. We follow the typical way of training diffusion policies, which optimizes $\psi$ by treating $p_{G_\theta}$ as the target action distribution (Wang et al., 2024),

$$\min_\psi \mathbb{E}_{x^k \sim q(x^k|x^0), x^0 = \text{stop-grad}(G_\theta(z)), z \sim \mathcal{N}(\mathbf{0}, \mathbf{I}), k \sim \mathcal{U}}[\lambda(k) \cdot ||\epsilon_\psi(x^k, k) - \epsilon_k||^2]. \quad (6)$$

Then we can obtain $s_{p_{\pi_\psi}}(\mathbf{A}_\theta^k)$ by applying $\epsilon_\psi$ to Equation (1). We approximate $s_{p_{G_\theta}}(\mathbf{A}_\theta^k)$ in Equation (5) with $s_{p_{\pi_\psi}}(\mathbf{A}_\theta^k)$. We iteratively update the generator parameters $\theta$ by Equation (5), and the generator score network parameter $\psi$ by Equation (6). The parameter of the prertrained diffusion policy $\phi$ is fixed throughout the training. During inference, we directly perform one-step sampling with Equation (3). We name our algorithm OneDP-S, where $S$ denotes the stochastic policy.

When we apply a deterministic action generator by omitting random noise $z$, such that $\mathbf{A}_\theta = G_\theta(\mathbf{O})$, the distribution $p_{G_\theta}$ becomes a Dirac delta function centered at $G_\theta(\mathbf{O})$, that is, $p_{G_\theta} = \delta_{G_\theta(\mathbf{O})}(\mathbf{A})$.

Consequently, $s_{p_{G_\theta}}(\mathbf{A}_\theta^k)$ can be explicitly solved as follows:

$$s_{p_{G_\theta}}(\mathbf{A}_\theta^k) = \nabla_{\mathbf{A}_\theta^k} \log p_\theta(\mathbf{A}_\theta^k) = \nabla_{\mathbf{A}_\theta^k} \log p_\theta(\mathbf{A}_\theta^k|\mathbf{A}_\theta) = -\frac{\boldsymbol{\epsilon}_k}{\sigma_k}; \mathbf{A}_\theta^k = \alpha_k \mathbf{A}_\theta + \sigma_k \boldsymbol{\epsilon}_k, \boldsymbol{\epsilon}_k \sim \mathcal{N}(\mathbf{0}, \boldsymbol{I}). \tag{7}$$

By incorporating Equation (7) into Equation (5), we can have a simplified loss function without the need of introducing the generator score network:

$$\nabla_\theta \mathbb{E}_{k\sim\mathcal{U}}[\mathcal{D}_{KL}(p_{G_\theta,k}||p_{\pi_\phi,k})] = \mathbb{E}_{\substack{\boldsymbol{z}\sim\mathcal{N}(\mathbf{0},\boldsymbol{I}),k\sim\mathcal{U} \\ \mathbf{A}_\theta = G_\theta(\boldsymbol{z},\mathbf{O}) \\ \mathbf{A}_\theta^k \sim q(\mathbf{A}_\theta^k|\mathbf{A}_\theta,k)}} \left[ \frac{w(k)}{\sigma_k}(\epsilon_\phi(\mathbf{A}_\theta^k,k)) - \epsilon_k)\nabla_\theta \mathbf{A}_\theta^k \right]. \tag{8}$$

We name this deterministic diffusion policy distillation OneDP-D. We illutrate our training pipeline in Figure 2, and summarize our algorithm training in Algorithm 1.

**Policy Discussion.** A stochastic policy, which encompasses deterministic policies, is more versatile and better suited to scenarios requiring exploration, potentially leading to better convergence at a global optimum (Haarnoja et al., 2018). In our case, OneDP-D simplifies the training process, though it may exhibit slightly weaker empirical performance. We offer a comprehensive comparison between OneDP-S and OneDP-D in Section 3.

**Distillation Discussion.** We discuss the benefits of optimizing the expectational reverse KL divergence. First, reverse KL divergence typically induces mode-seeking behavior, which has been shown to improve empirical performance in offline RL (Chen et al., 2023b). Therefore, we anticipate that reverse KL-based distillation offers similar advantages for robotic tasks. Second, as demonstrated by Wang et al. (2023), optimizing JSD, a combination of KLs, between diffused action samples provides stronger performance when dealing with distributions with misaligned supports. This aligns with our approach of performing KL optimization over the diffused distribution.

---

**Algorithm 1** OneDP Training

1: **Inputs:** action generator $G_\theta$, generator score network $\pi_\psi$, pre-trained diffusion policy $\pi_\phi$.
2: **Initializaiton** $G_\theta \leftarrow \pi_\phi, \pi_\psi \leftarrow \pi_\phi$.
3: **while** not converged **do**
4:     Sample $\mathbf{A}_\theta = G_\theta(\boldsymbol{z},\mathbf{O}), \boldsymbol{z} \sim \mathcal{N}(\mathbf{0},\boldsymbol{I})$.
5:     Diffuse $\mathbf{A}_\theta^k = \alpha_k \mathbf{A}_\theta + \sigma_k \boldsymbol{\epsilon}_k, \boldsymbol{\epsilon}_k \sim \mathcal{N}(\mathbf{0},\boldsymbol{I})$.
6:     **if** OneDP-S **then**
7:         Update $\psi$ by Equation (6)
8:         Update $\theta$ by Equation (5)
9:     **else if** OneDP-D **then**
10:        Update $\theta$ by Equation (8)
11:    **end if**
12: **end while**

---

### 2.3 IMPLEMENTATION DETAILS

**Diffusion Policy.** Following Chi et al. (2023), we construct a diffusion policy using a 1D temporal convolutional neural network (CNN) (Janner et al., 2022) based U-Net and a standard ResNet18 (without pre-training) (He et al., 2016) as the vision encoder. We implement the diffusion policy with two noise scheduling methods: DDPM (Ho et al., 2020) and EDM (Karras et al., 2022). We use $\epsilon$ noise prediction for discrete-time (100 steps) diffusion and $x^0$ prediction for continuous-time diffusion, respectively. The EDM scheduling is essential for Consistency Policy (Prasad et al., 2024) due to the use of CTM (Kim et al., 2023a). For DDPM, we set $\lambda(k) = 1$ and use the original SDE and DDIM (Song et al., 2020a) sampling. For EDM, we use the default $\lambda(k) = \frac{\sigma_k^2 + \sigma_d^2}{(\sigma_k \sigma_d)^2}$ with $\sigma_d = 0.5$. We use the second-order EDM sampler, which requires two neural network forwards per discretized step in the ODE.

**Distillation.** We warm-start both the stochastic and deterministic action generator $G_\theta$, and the generator score network, $\epsilon_\psi$, by duplicating the neural-network structure and weights from the pre-trained diffusion policy, aligning with strategies from Luo et al. (2024); Yin et al. (2024); Xu et al. (2024). The inputs of $G_\theta$ include pure noise, a fixed time embedding (an initial timestep for DDPM or initial sigma value for EDM), and observations $\mathbf{O}$. The outputs of $G_\theta$ are formulated as direct action predictions. Following DreamFusion (Poole et al., 2022), we set $w(k) = \sigma_k^2$. In the discrete-time domain, distillation occurs over [2, 95] diffusion timesteps to avoid edge cases. In continuous-time, we employ the same log-normal noise scheduling as EDM (Karras et al., 2022) used during distillation. The generators operate at a learning rate of $1 \times 10^{-6}$, while the generator

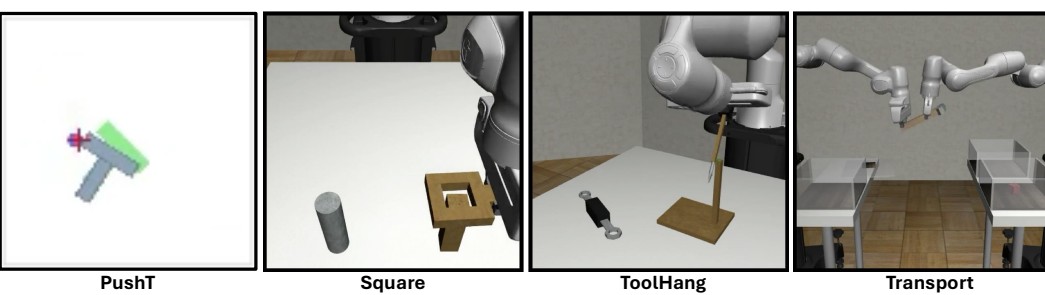

Figure 3: **Simulation tasks.** We evaluate our method against baselines on the single-robot tasks: PushT, Square, and ToolHang, as well as a dual-robot task Transport. Task difficulty increases from left to right.

Table 1: **Robomimic Benchmark Performance (Visual Policy) in DDPM**. We compare our proposed OneDP-D and OneDP-S, with DP under the default DDPM scheduling. We report the mean and standard deviation of success rates across 5 different training runs, each evaluated with 100 distinct environment initializations. Details of the evaluation procedure can be found in Section 3.1. Our results demonstrate that OneDP not only matches but can even outperform the pre-trained DP, achieving this with just one-step generation, resulting in an order of magnitude speed-up.

| Method | Epochs | NFE | PushT | Square-mh | Square-ph | ToolHang-ph | Transport-mh | Transport-ph | Avg |
|---|---|---|---|---|---|---|---|---|---|
| DP (DDPM) | 1000 | 100 | **0.863 ± 0.040** | 0.846 ± 0.023 | **0.926 ± 0.023** | 0.822 ± 0.016 | 0.620 ± 0.049 | 0.896 ± 0.032 | 0.829 |
| DP (DDIM) | 1000 | 10 | 0.823 ± 0.023 | 0.850 ± 0.013 | 0.918 ± 0.009 | 0.828 ± 0.016 | 0.688 ± 0.020 | 0.908 ± 0.011 | 0.836 |
|  | 1000 | 1 | 0.000 ± 0.000 | 0.000 ± 0.000 | 0.000 ± 0.000 | 0.000 ± 0.000 | 0.000 ± 0.000 | 0.000 ± 0.000 | 0.000 |
| OneDP-D | 20 | 1 | 0.802 ± 0.057 | 0.846 ± 0.028 | 0.926 ± 0.011 | 0.808 ± 0.046 | 0.676 ± 0.029 | 0.896 ± 0.013 | 0.826 |
| OneDP-S | 20 | 1 | 0.816 ± 0.058 | **0.864 ± 0.042** | **0.926 ± 0.018** | **0.850 ± 0.033** | **0.690 ± 0.024** | **0.914 ± 0.021** | **0.843** |

Table 2: **Robomimic Benchmark Performance (Visual Policy) in EDM**. We compare our proposed OneDP with CP under the EDM scheduling. EDM scheduling is required in CP to satisfy boundary conditions. We follow our evaluation metric and report similar values as in Table 1. We also ablate Diffusion Policy with 1, 10 and 18 ODE steps, which utilizes 1, 19 and 35 NFE in EDM sampling.

| Method | Epochs | NFE | PushT | Square-mh | Square-ph | ToolHang-ph | Transport-mh | Transport-ph | Avg |
|---|---|---|---|---|---|---|---|---|---|
| DP (EDM) | 1000 | 35 | **0.861 ± 0.030** | 0.810 ± 0.026 | 0.898 ± 0.033 | **0.828 ± 0.019** | 0.684 ± 0.019 | 0.890 ± 0.012 | 0.829 |
|  | 1000 | 19 | 0.851 ± 0.012 | **0.828 ± 0.015** | 0.880 ± 0.014 | 0.794 ± 0.012 | 0.692 ± 0.009 | 0.860 ± 0.013 | 0.818 |
|  | 1000 | 1 | 0.000 ± 0.000 | 0.000 ± 0.000 | 0.000 ± 0.000 | 0.000 ± 0.000 | 0.000 ± 0.000 | 0.000 ± 0.000 | 0.000 |
| CP | 20 | 1 | 0.595 ± 0.141 | 0.120 ± 0.165 | 0.238 ± 0.219 | 0.238 ± 0.163 | 0.140 ± 0.148 | 0.174 ± 0.257 | 0.251 |
| CP | 450 | 1 | 0.828 ± 0.055 | 0.646 ± 0.047 | 0.776 ± 0.055 | 0.650 ± 0.046 | 0.378 ± 0.091 | 0.754 ± 0.120 | 0.672 |
| CP | 450 | 3 | 0.839 ± 0.037 | 0.710 ± 0.018 | 0.874 ± 0.022 | 0.626 ± 0.041 | 0.374 ± 0.051 | 0.848 ± 0.028 | 0.712 |
| OneDP-D | 20 | 1 | 0.829 ± 0.052 | 0.776 ± 0.023 | 0.902 ± 0.040 | 0.762 ± 0.056 | 0.705 ± 0.038 | 0.898 ± 0.019 | 0.812 |
| OneDP-S | 20 | 1 | 0.841 ± 0.042 | 0.774 ± 0.033 | **0.910 ± 0.041** | 0.824 ± 0.039 | **0.722 ± 0.025** | **0.910 ± 0.027** | **0.830** |

score network is accelerated to a learning rate of $2 \times 10^{-5}$. Vision encoders are also actively trained during the distillation process.

# 3 EXPERIMENTS

We evaluate OneDP on a wide variety of tasks in both simulated and real environments. In the following sections, we first report the evaluation results in simulation across six tasks that include different complexity levels. Then we demonstrate the results in the real environment by deploying OneDP in the real world with a Franka robot arm for object pick-and-place tasks and a coffee-machine manipulation task. We compare our method with the pre-trained backbone Diffusion Policy (Chi et al., 2023) (DP) and related distillation baseline Consistency Policy (Prasad et al., 2024) (CP). We also report the ablation study results in Appendix C to present more detailed analyses on our method and discuss the effect of different design choices.

## 3.1 Simulation Experiments

**Datasets.** Robomimic. Proposed in (Mandlekar et al., 2021), Robomimic is a large-scale benchmark for robotic manipulation tasks. The original benchmark consists of five tasks: Lift, Can, Square, Transport, and Tool Hang. We find that the the performance of state-of-the-art methods was already saturated on two easy tasks Lift and Can, and therefore only conduct the evaluation on the harder tasks Square, Transport and Tool Hang. For each of these tasks, the benchmark provides two variants of human demonstrations: proficient human (PH) demonstrations and mixed proficient/non-proficient human (MH) demonstrations. PushT. Adapted from IBC (Florence et al., 2022), Chi et al. (2023) introduced the PushT task, which involves pushing a T-shaped block into a fixed target using a circular end-effector. A dataset of 200 expert demonstrations is provided with RGB image observations.

**Experiment Setup.** We pretrain the DP model for 1000 epochs on each benchmark under both DDPM (Ho et al., 2020) and EDM (Karras et al., 2022) noise scheduling. Note EDM noise scheduling is a requirement for CP (Prasad et al., 2024) to satisfy diffusion boundary conditions. Subsequently, we train OneDP for 20 epochs and the baseline CP for 450 epochs until convergence. During evaluation, we observe significant variance in evaluating success rates with different environment initializations. We present average success rates across 5 training seeds and 100 different initial conditions (500 in total). We report the peak success rate for each method during training, corresponding to the peak points of the curves in Figure 4. The metric for most tasks is the success rate, except for PushT, which is evaluated using the coverage of the target area.

Table 1 presents the results of OneDP compared with DP under the default DDPM setting. For DP, we report the average success rate using DDPM sampling with 100 timesteps, as well as the accelerated DDIM sampling with 1 and 10 timesteps. Notably, DP fails to generate reasonable actions with single-step generation, yielding a 0% success rate for all tasks. DP with 10 steps under DDIM slightly outperforms DP under DDPM. However, OneDP demonstrates the highest average success rate with single-step generation across the six tasks, with the stochastic variant OneDP-S surpassing the deterministic OneDP-D. This superior performance of OneDP-S aligns with our discussion in Section 2.2, suggesting that stochastic policies generally perform better in complex environments. Interestingly, OneDP-S even slightly outperforms the pre-trained DP, which is not unprecedented, as shown in cases of image distillation (Zhou et al., 2024) and offline RL (Chen et al., 2023b). We attribute this to the fact that iterative sampling may introduce subtle cumulative errors during the denoising process, whereas single-step sampling avoids this issue by jumping directly from the end to the start of the reverse diffusion chain.

In Table 2, we report a similar comparison under the EDM setting, including CP. We report DP under the same 1 and 10 DDIM steps, and 100 DDPM steps, which correspond to 1, 19, and 35 number of function evaluations (NFE) in EDM due to second-order ODE sampling. OneDP-S outperforms the baseline CP with single-step and its default best setting of 3-step chain generation. Under EDM, OneDP-S matches the average success rate of the pre-trained DP, while OneDP-D performs slightly worse. We also observe that CP converges much more slowly compared to OneDP, as shown in Figure 4. This slower convergence is likely because CP, based on CTM, does not involve the auxiliary discriminator training that is used to enhance distillation performance in CTM.

## 3.2 Real World Experiments

We design four tasks to evaluate the real-world performance of OneDP, including three common tasks where the robot picks and places objects at designated locations, referred to as `pnp`, and one challenging task where the robot learns to manipulate a coffee machine, called `coffee`. Figure 5 shows the experimental setup, with the first row illustrating the `pnp` tasks and the second row depicting the `coffee` task. We introduce the data collection process and the evaluation setup in the following section and provide more details in Appendix A.

**`pnp` Tasks.** This task requires the robot to pick an object from the table and put it in a box. We design three variants of this task: `pnp-milk`, `pnp-anything` and `pnp-milk-move`. In `pnp-milk`, the object is always the same milk box. In `pnp-anything`, we expand the target to 11 different objects as shown in Figure 8. For `pnp-milk-move`, we involve human interference to create a dynamic environment. Whenever the robot gripper attempts to grasp the milk box, we move it away, following the trajectory as shown in Figure 9. We collect 100 demonstrations each for the `pnp-milk` and `pnp-anything` tasks. Separate models are trained for both tasks, with the

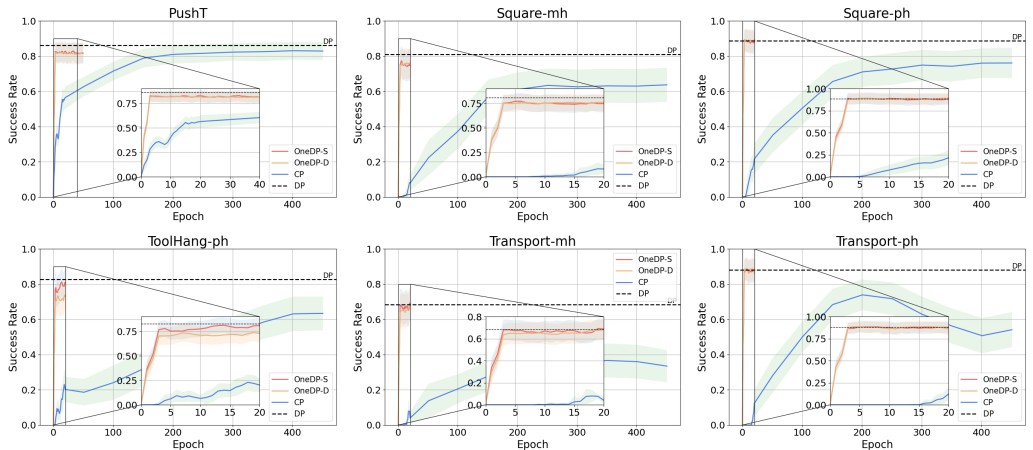

Figure 4: **Convergence Comparison.** We show our method OneDP converges $20\times$ faster than the baseline method Consistency Policy (CP) under EDM setting.

`pnp-anything` model utilizing all 200 demonstrations. The `pnp-milk-move` task is evaluated using the checkpoint from the `pnp-anything` model.

**`Coffee` Task.** This task requires the robot to operate a coffee machine. It involves the following steps: (1) picking up the coffee pod, (2) placing the coffee pod in the pod holder on the coffee machine, and (3) closing the lid of the coffee machine. This task is more challenging since it involves more steps and requires the robot to insert the pod in the holder accurately. We collect 100 human demonstrations for this task. We train one specific model for this task.

**Evaluation.** We evaluate the success rate and task completion time from 20 predetermined initial positions for the `pnp-milk`, `pnp-anything`, and `coffee` tasks, as well as 10 motion trajectories for the `pnp-milk-move` task. The left side of Figure 7 shows the setup of the robot, destination box, and coffee machine, with 20 fixed initialization points. Figure 9 shows the 10 trajectories for evaluating `pnp-milk-move`. Details of the evaluation are provided in Appendix A. For DP, we follow Chi et al. (2023) to use DDIM (10 steps) to accelerate the real-world experiment.

We compare OneDP against the DP backbone in real-world experiments, focusing on three key aspects: success rate, responsiveness, and time efficiency. Table 3 demonstrates that OneDP consistently outperforms DP across all tasks, with the most significant improvement seen in `pnp-milk-move`. This task demands rapid adaptation to dynamic environmental changes, particularly due to sudden human interference. The wall-clock time for action generation is reported in Table 5. The slow action generation of DP hinders its ability to track the moving milk box effectively, often losing control when the box moves out of its visual range, as it is still predicting actions based on outdated information. In contrast, OneDP generates actions quickly, allowing it to instantly follow the box's movement, achieving a 100% success rate in this dynamic task. OneDP-S slightly outperforms OneDP-D, aligning with the observations from the simulation experiments.

Additionally, we measure the task completion time for successful evaluation rollouts across all algorithms. As shown in Table 4, OneDP completes tasks faster than DP. Both OneDP-S and OneDP-D exhibit similarly-rapid task completion times. The quick action prediction of OneDP reduces hesitation during robot arm movements, particularly when the arm camera's viewpoint changes abruptly. This leads to significant improvements in task completion speed. In Figure 7, we present a heatmap for illustrating the task completion times; lighter colors indicate faster completion times, while dark red demonstrates failure cases. Overall, OneDP completes tasks more efficiently across most locations. Although all three algorithms encounter failures in some corner cases for the `coffee` task, OneDP-S shows fewer failures.

## 4 RELATED WORK

**Diffusion Models.** Diffusion models have emerged as a powerful framework for modeling complex data distributions and have achieved groundbreaking performance across various tasks involving

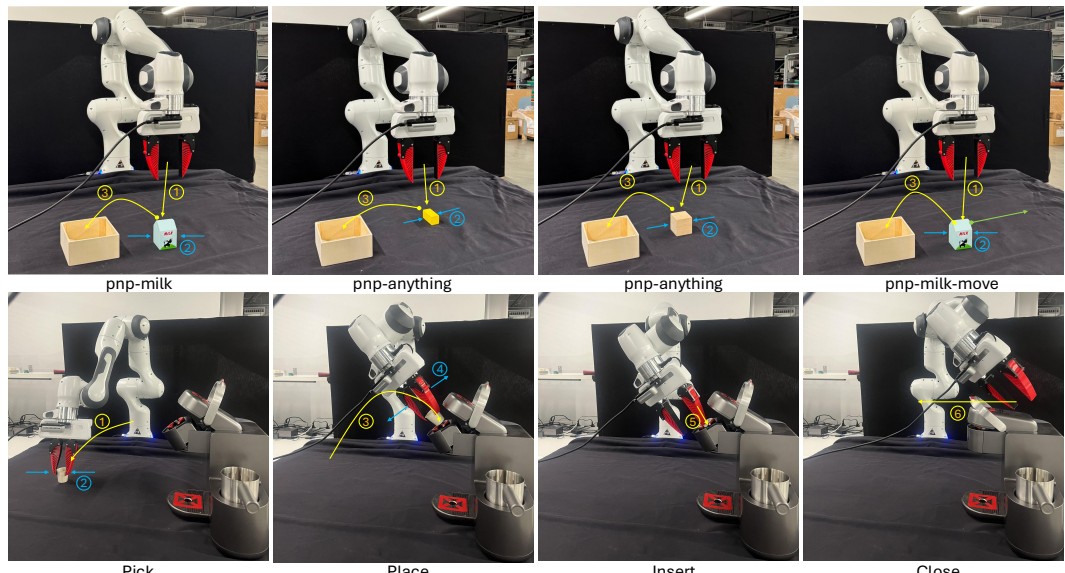

Figure 5: **Real-World Experiment Illustration.** In the first row, we display the setup for the pick-and-place experiments, featuring three tasks: `pnp-milk`, `pnp-anything`, and `pnp-milk-move`. In total, `pnp-anything` handles around 10 random objects as shown in Figure 8. The second row illustrates the procedure for the more challenging `coffee` task, where the Franka arm is tasked with locating the coffee cup, precisely positioning it in the machine's cup holder, inserting it, and finally closing the machine's lid.

Table 3: **Success Rate of Real-world Experiments**. We evaluate the performance of our proposed OneDP-D and OneDP-S against the baseline Diffusion Policy in real-world robotic manipulation tasks. The baseline Diffusion Policy was trained for 1000 epochs to ensure convergence, whereas our distilled models were trained for 100 epochs. We do not select checkpoints; only the final checkpoint is used for evaluation. Performance is assessed over 20 predetermined rounds, and we report the average success rate.

| Method | Epochs | NFE | pnp-milk | pnp-anything | pnp-milk-move | coffee | Avg |
|---|---|---|---|---|---|---|---|
| DP(DDIM) | 1000 | 10 | **1.00** | 0.95 | 0.80 | 0.80 | 0.83 |
| OneDP-D | 100 | 1 | **1.00** | **1.00** | **1.00** | 0.80 | 0.95 |
| OneDP-S | 100 | 1 | **1.00** | **1.00** | **1.00** | **0.90** | **0.98** |

Table 4: **Time Efficiency of Real-world Experiments**. We present the completion times for each algorithm as recorded in Table 3. For a fair comparison, we report the average completion time (in seconds) for each algorithm across evaluation rounds where all algorithms succeeded. Specifically, the tasks `pnp-milk`, `pnp-anything`, `pnp-milk-move`, and `coffee` were averaged over 18, 15, 8, and 13 respective rounds. These times indicate how quickly each algorithm responds and completes tasks in a real-world environment.

| Method | Epochs | NFE | pnp-milk | pnp-anything | pnp-milk-move | coffee | Avg |
|---|---|---|---|---|---|---|---|
| DP(DDIM) | 1000 | 10 | 29.74 | 26.03 | 34.75 | 54.92 | 36.36 |
| OneDP-D | 100 | 1 | 23.21 | 22.93 | 28.73 | 33.13 | 27.00 |
| OneDP-S | 100 | 1 | **22.69** | **22.62** | **28.15** | **29.78** | **25.81** |

generative modeling (Ho et al., 2020; Karras et al., 2022). They operate by transforming data into Gaussian noise through a diffusion process and subsequently learning to reverse this process via iterative denoising. Diffusion models have been successfully applied to a wide range of domains, including image, video, and audio generation Saharia et al. (2022); Ramesh et al. (2022); Balaji et al. (2022); Chen et al. (2023a); Ho et al. (2022); Popov et al. (2021); Kong et al. (2020), reinforcement learning (Janner et al., 2022; Wang et al., 2022; Psenka et al., 2023) and robotics (Ajay et al., 2022; Urain et al., 2023; Chi et al., 2023).

Table 5: **Real-world inference speeds.** We report the wall clock times for each policy in real-world scenarios. The action generation process consists of two parts: observation encoding (OE) and action prediction by each method. All measurements were taken using a local NVIDIA V100 GPU, with the same neural network size for each method. The policy frequencies, shown in Figure 1, are based on the values from this table.

|          | OE | DDPM (100 steps) | DDIM (10 steps) | OneDP (1 step) |
|----------|----|------------------|-----------------|----------------|
| Time (ms) | 9  | 660              | 66              | 7              |
| NFE      | 1  | 100              | 10              | 1              |

**Diffusion Policies.** Diffusion models have shown promising results as policy representations for control tasks. Janner et al. (2022) introduced a trajectory-level diffusion model that predicts all timesteps of a plan simultaneously by denoising two-dimensional arrays of state and action pairs. Wang et al. (2022) proposed Diffusion Q-learning, which leverages a conditional diffusion model to represent the policy in offline reinforcement learning. An action-space diffusion model is trained to generate actions conditioned on the states. Similarly, Chi et al. (2023) used a conditional diffusion model in the robot action space to represent the visuomotor policy and demonstrated a significant performance boost in imitation learning for various robotics tasks. Ze et al. (2024) further incorporated the power of a compact 3D visual representations to improve diffusion policies in robotics.

**Diffusion Distillations.** Although diffusion models are powerful, their iterative denoising process makes them inherently slow in generation, which poses challenges for time-sensitive applications like robotics and real-time control. Motivated by the need to accelerate diffusion models, diffusion distillation has become an active research topic in image generation. Diffusion distillation aims to train a student model that can generate samples with fewer denoising steps by distilling knowledge from a pre-trained teacher model (Salimans & Ho, 2022; Luhman & Luhman, 2021; Zheng et al., 2023; Song et al., 2023; Kim et al., 2023b). Salimans & Ho (2022) proposed a method to distill a teacher model into a new model that takes half the number of sampling steps, which can be further reduced by progressively applying this procedure. Song et al. (2023) introduced consistency models that enable fewer step sampling by enforcing self-consistency of the ODE trajectories. CTM (Kim et al., 2023b) improved consistency models and provided the flexibility to trade-off quality and speed. (Luo et al., 2024; Yin et al., 2024) leverage the success of stochastic distillation sampling (Poole et al., 2022) in text-to-3D and proposes KL-based score distillation for image generation. Beyond KL, Zhou et al. (2024) proposes the SiD distillation technique derived from Fisher Divergence. However, leveraging diffusion distillation to accelerate diffusion policies for robotics remains an underexplored and pressing challenge, particularly for real-time control applications. Consistency Policy (Prasad et al., 2024) explored applying CTM to reduce the number of denoising steps and accelerate inference of the diffusion policies. It simplifies the original CTM training by ignoring the adversarial auxiliary loss. While this approach achieves a considerable speed-up, it leads to performance degradation compared to pre-trained models, and its complex training process and slow convergence present challenges for robotics applications. In contrast, OneDP employs expectational reverse KL optimization to distill a powerful one-step action generator, achieving comparable or higher success rates than the original diffusion policy, while converging $20\times$ faster.

## 5 CONCLUSION

In this paper, we introduced the One-Step Diffusion Policy (OneDP) through advanced diffusion distillation techniques. We enhanced the slow, iterative action prediction process of Diffusion Policy by reducing it to a single-step process, dramatically decreasing action inference time and enabling the robot to respond quickly to environmental changes. Through extensive simulation and real-world experiments, we demonstrate that OneDP not only achieves a slightly higher success rate, but also responds quickly and effectively to environmental interference. The rapid action prediction further allows the robot to complete tasks more efficiently.

However, this work has some limitations. In the experiments, we did not test OneDP on long-horizon real-world tasks. Furthermore, in the real-world experiments, we limited the robot's operation frequency to 20 Hz for controlling stability, which underutilized OneDP 's full potential. Additionally,

the KL-based distillation method may not be the optimal choice for distribution matching, and introducing a discriminator term could potentially improve distillation performance.

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

## A  REAL-WORLD EXPERIMENT SETUP

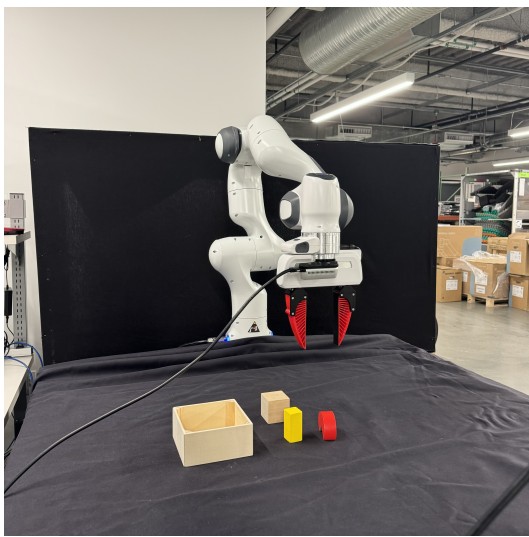

Figure 6: Real-world Experiment Setup

**Robot Setup.**    The physical robot setup consists of a Franka Panda robot arm, a front-view Intel RealSense D415 RGB-D camera, and a wrist-mounted Intel RealSense D435 RGB-D camera. The RGB image resolution was set to 120x160. The depth image is not used in our experiments.

**Teleoperation.**    Demonstration data for the real robot tasks was collected using a phone-based teleoperation system (Mandlekar et al., 2018; 2019).

**Data Collection.**    We collect 100 demonstrations for each task separately: `pnp-milk`, `pnp-anything`, and `coffee`. In `pnp-milk`, the target object is always the milk box, and the task involves picking up the milk box from various random locations and placing it into a designated target box at a fixed location. For `pnp-anything`, we extend the set of target objects to 11 different items, as shown in Figure 8, with the target box location randomized vertically. In the `coffee` task, the coffee cup is randomly placed, and the robot is required to pick it up, insert it into the coffee machine, and close the lid.

The area and location for each task are illustrated in the left column of Figure 7. During data collection, target objects are randomly positioned within the blue area; the grid is used for evaluation, as described in the next section. For the `pnp` tasks, the blue area is a rectangle measuring 23 cm in height and 20 cm in width, while the target box is a square with a side length of 13 cm. In the `coffee` task, the blue area is slightly smaller, measuring 18 cm in height and 20 cm in width.

Table 6: **Real-world experiment demonstrations.** In total we collect 300 demonstrations, with 100 demonstrations for each task.

|  | pnp-milk | pnp-anything | coffee |
|---|---|---|---|
| Demos | 100 | 100 | 100 |

**Evaluation.**    To ensure a fair comparison between OneDP and all baseline methods, we standardize the evaluation process. For the `pnp-milk`, `pnp-anything`, and `coffee` tasks, we evaluate each method according to the grid order shown in Figure 7. The target object is placed at the center of the grid to ensure consistent initial conditions across evaluations. For task `pnp-anything`, the picked object also follows the order shown in Figure 8. For the dynamic environment task `pnp-milk-move`, we introduce human interference during the evaluation. Whenever the robot

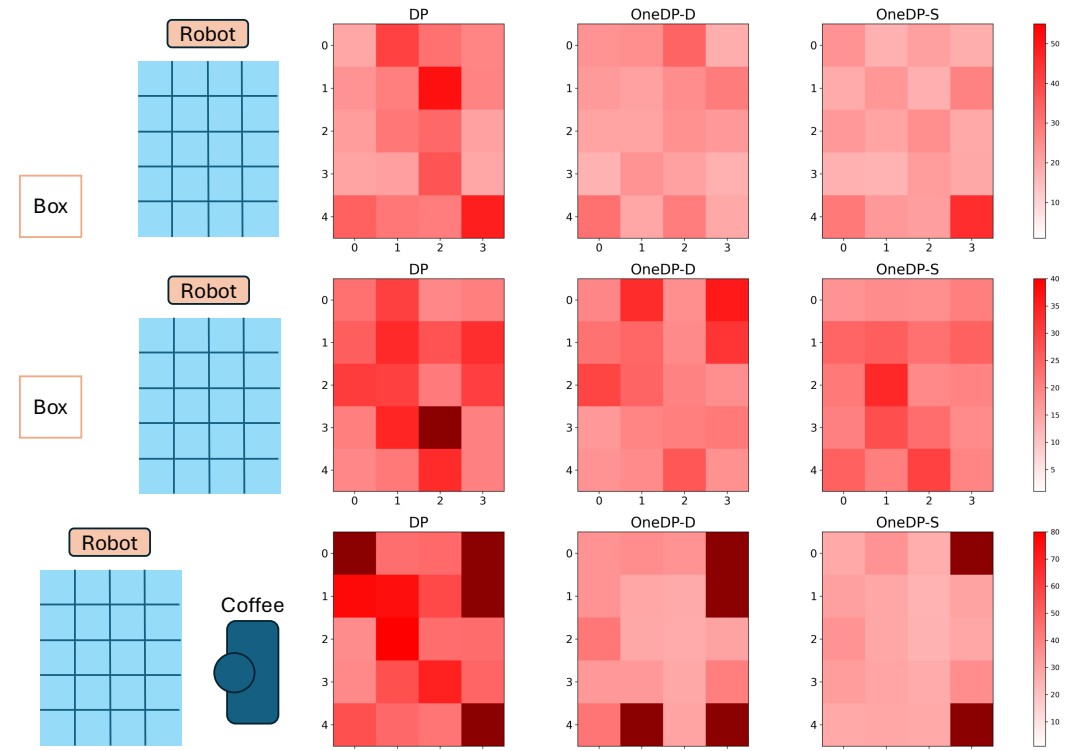

Figure 7: **Real-World Comparison Illustration.** We present the time taken by each algorithm to complete tasks from a specific starting point in colors. A color map on the right side ranges from white to red indicating the time in seconds. Dark red signifies that the algorithm failed at that location. The three rows represent tasks `pnp-milk`, `pnp-anything`, `coffee`. Details of the evaluation of `pnp-anything` can be found in Figure 8.

gripper attempts to grasp the target milk box, we manually move it away along the trajectory depicted in Figure 9. Although we aim to maintain consistent conditions during each evaluation, the exact nature of human interference cannot be guaranteed. Some trajectories involve a single instance of interference, while others may involve two consecutive human movements.

The original DDPM sampling in Diffusion Policy is too slow for real-world experiments. To speed up the evaluation, we follow (Chi et al., 2023) and use DDIM with 10 steps. For OneDP, we use single-step generation. In real-world experiments, we do not select intermediate checkpoints but use the final checkpoint after training for each method.

We record both the success rates and completion times, reporting their mean values. For `pnp-milk-move`, evaluations are conducted over 10 trajectories, while for the other tasks, results are obtained from 20 grid points. In Figure 7, we present a heatmap to visualize task completion times, where lighter colors represent faster completions and dark red indicates failure cases. Overall, OneDP completes tasks more efficiently across most locations. While all three algorithms experience failures in certain corner cases for the `coffee` task, OneDP-S demonstrates fewer failures.

## B  TRAINING DETAILS

We follow the CNN-based neural network architecture and observation encoder design from Chi et al. (2023). For simulation experiments, we use a 256-million-parameter version for DDPM and a 67-million-parameter version for EDM, as the smaller EDM network performs slightly better. In real-world experiments, we also use the 67-million-parameter version. Additionally, we adopt the action chunking idea from Chi et al. (2023) and Zhao et al. (2023), using 16 actions per chunk for prediction, and utilize two observations for vision encoding.

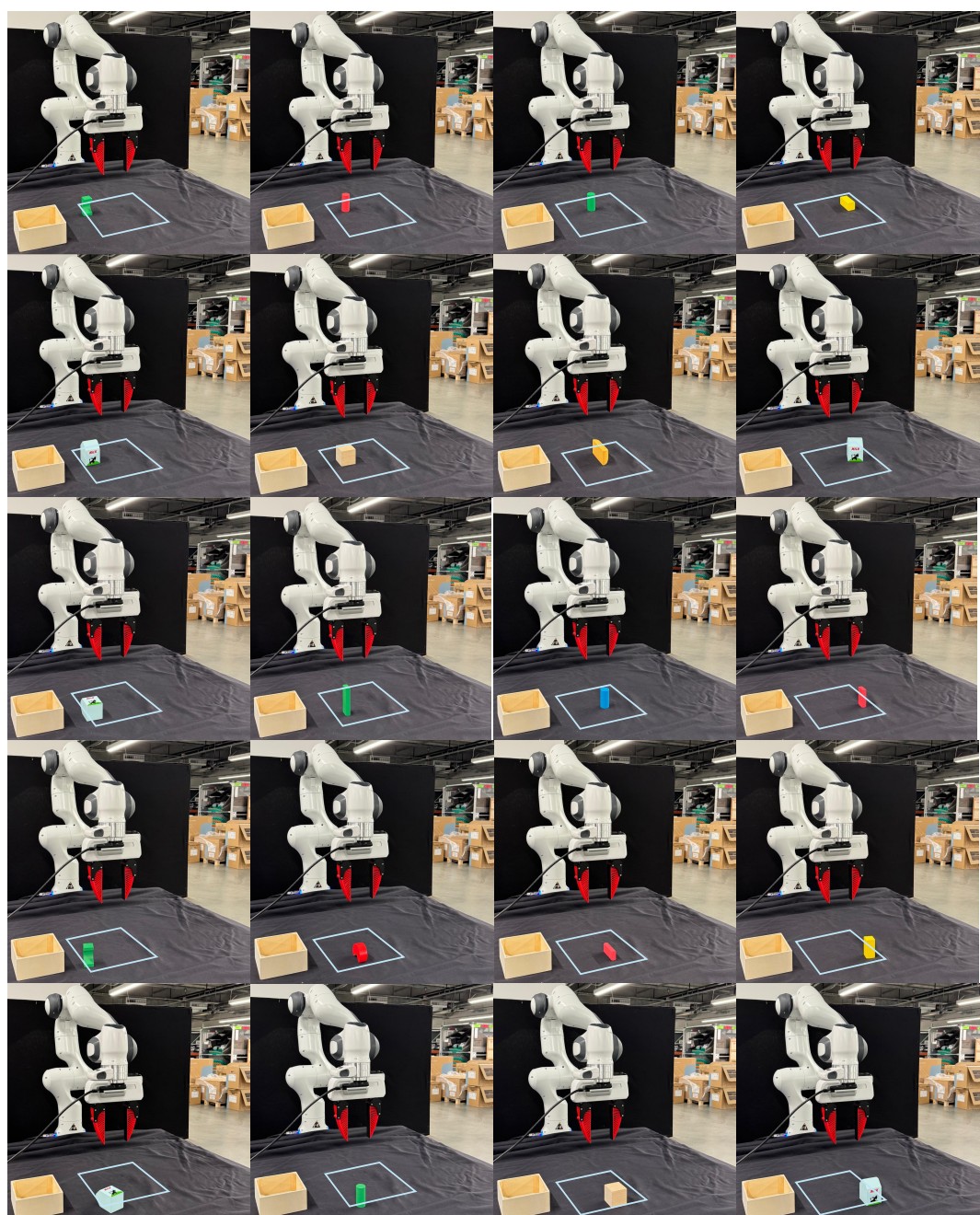

Figure 8: Evaluation setup for `pnp-anything`.

We first train DP for 1000 epochs in both simulation and real-world experiments with a default learning rate of 1e-4 and weight decay of 1e-6. We then perform distillation using the pre-trained checkpoints, distilling for 20 epochs in simulation and 100 epochs in real-world experiments.

For distillation, we warm-start both the stochastic and deterministic action generators, $G_\theta$, and the generator score network, $\epsilon_\psi$, by duplicating the network structure and weights from the pre-trained diffusion-policy checkpoints. Since the generator network is initialized from a denoising network, a timestep input is required, as this was part of the original input. We fix the timestep at 65 for discrete diffusion and choose $\sigma = 2.5$ for continuous EDM diffusion. The generator learning rate is set to 1e-6. We find these hyperparameters to be stable without causing significant performance variation.

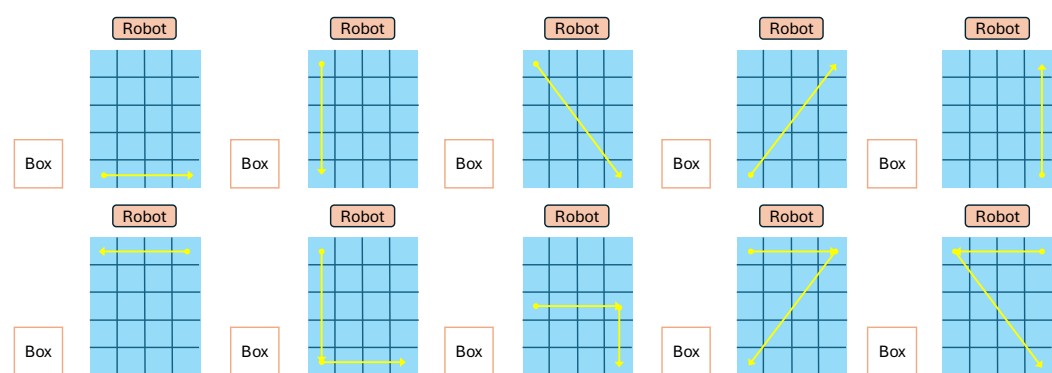

Figure 9: Evaluation trajectories for `pnp-milk-move`. The box is always on the left-hand side of the tested blue area.

We provide an ablation study that focuses primarily on the generator score network's learning rate and optimizer settings in Appendix C. We provide the hyperparameter details in Table 7.

| Hyperparameters | Values |
| --- | --- |
| generator learning rate | lr=1e-6 |
| generator score network learning rate | lr=2e-5 |
| generator optimizer | Adam([0.0, 0.999]) |
| generator score network optimizer | Adam([0.0, 0.999]) |
| action chunk size | n=16 |
| number of observations | n=2 |
| discrete diffusion init timestep | $t_{\text{init}}$=65 |
| discrete diffusion distillation $t$ range | [2, 95] |
| continuous diffusion init sigma | $\sigma = 2.5$ |

Table 7: Hyperparameters

## C  ABLATION STUDY

As shown in the first panel of Figure 10, we explore a range of learning rates for the generator score network in the grid [1e-6, 1e-5, 2e-5, 3e-5, 4e-5] and find 2e-5 to be optimal in most cases. A higher learning rate for the score network compared to the generator ensures that the score network keeps pace with the generator's distribution updates during training. In the second panel, we search for the best optimizer settings, finding that setting $\beta_1$ to 0 for both the generator and the generator score network optimizers is effective. This approach, commonly used in GANs, allows the two networks to evolve together more quickly.

## D  DETAILED PRELIMINARIES

Diffusion models are robust generative models utilized across various domains (Ho et al., 2020; Sohl-Dickstein et al., 2015; Song et al., 2020b). They operate by establishing a forward diffusion process that incrementally transforms the data distribution into a known noise distribution, such as standard Gaussian noise. A probabilistic model is then trained to methodically reverse this diffusion process, enabling the generation of data samples from pure noise.

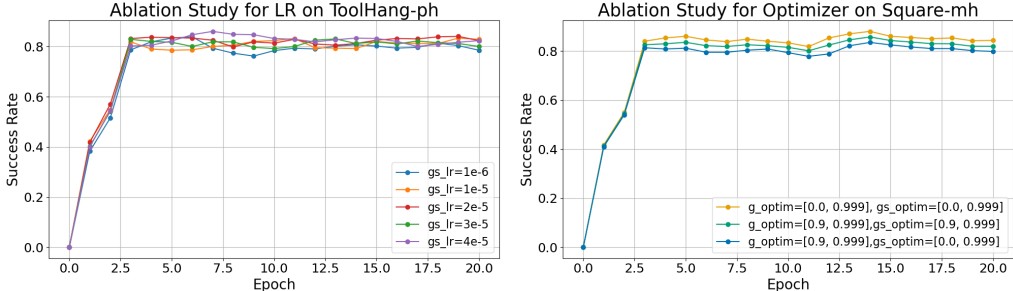

Figure 10: Ablation studies on the learning rate of the generator score network and optimizer settings.

Suppose the data distribution is $p(\boldsymbol{x})$. The forward diffusion process is conducted by gradually adding Gaussian noise to samples $\boldsymbol{x}^0 \sim p(\boldsymbol{x})$ as follows,

$$\boldsymbol{x}^k = \alpha_k \boldsymbol{x}^0 + \sigma_k \boldsymbol{\epsilon}_k, \boldsymbol{\epsilon}_k \sim \mathcal{N}(\boldsymbol{0}, \boldsymbol{I}); \quad q(\boldsymbol{x}^k|\boldsymbol{x}^0) := \mathcal{N}(\alpha_k \boldsymbol{x}^0, \sigma_k^2 \boldsymbol{I})$$

where $\alpha_k$ and $\sigma_k$ are parameters manually designed to vary according to different noise scheduling strategies. DDPM (Ho et al., 2020) is a discrete-time diffusion model with $k \in \{1, \ldots, K\}$. It can be easily extended to continuous-time diffusion from the score-based generative model perspective (Song et al., 2020b; Karras et al., 2022) with $k \in [0, 1]$. With sufficient amount of noise added, $\boldsymbol{x}^K \simeq \mathcal{N}(\boldsymbol{0}, \boldsymbol{I})$. Ho et al. (2020) propose to reverse the diffusion process and iteratively reconstruct the original sample $\boldsymbol{x}^0$ by training a neural network $\epsilon_\theta(\boldsymbol{x}^k, k)$ to predict the noise $\boldsymbol{\epsilon}_k$ added at each forward diffusion step (epsilon prediction). With reparameterization $\boldsymbol{\epsilon}_k = (\boldsymbol{x}^k - \alpha_k \boldsymbol{x}^0)/\sigma_k$, the diffusion model could also be formulated as a $\boldsymbol{x}^0$-prediction process $x_\theta(\boldsymbol{x}^k, k)$ (Karras et al., 2022; Xiao et al., 2021). We use epsilon prediction $\epsilon_\theta$ in our derivation. The diffusion model is trained with the denoising score matching loss (Ho et al., 2020),

$$\min_\theta \mathbb{E}_{\boldsymbol{x}^k \sim q(\boldsymbol{x}^k|\boldsymbol{x}^0), \boldsymbol{x}^0 \sim p(\boldsymbol{x}), k \sim \mathcal{U}}[\lambda(k) \cdot ||\epsilon_\theta(\boldsymbol{x}^k, k) - \boldsymbol{\epsilon}_k||^2]$$

where $\mathcal{U}$ is a uniform distribution over the $k$ space, and $\lambda(k)$ is a noise-ratio re-weighting function. With a trained diffusion model, we could sample $\boldsymbol{x}^0$ by reversing the diffusion chain, which involves discretizing the ODE (Song et al., 2020b) as follows:

$$d\boldsymbol{x}^k = \left[ f(k)\boldsymbol{x}^k - \frac{1}{2}g^2(k)\nabla_{\boldsymbol{x}_k} \log q(\boldsymbol{x}^k) \right] dk \tag{9}$$

where $f(k) = \frac{d \log \alpha_k}{dk}$ and $g^2(k) = \frac{d\sigma_k^2}{dk} - 2\frac{d \log \alpha_k}{dk}\sigma_k^2$. The unknown score $\nabla_{\boldsymbol{x}_k} \log q(\boldsymbol{x}^k)$ could be estimated as follows:

$$s(\boldsymbol{x}^k) = \nabla_{\boldsymbol{x}_k} \log q(\boldsymbol{x}^k) = -\frac{\epsilon^*(\boldsymbol{x}^k, k)}{\sigma_k} \approx -\frac{\epsilon_\theta(\boldsymbol{x}^k, k)}{\sigma_k},$$

where $\epsilon^*(\boldsymbol{x}^k, k)$ is the true noise added at time $k$, and we let $s_\theta(\boldsymbol{x}^k) = -\frac{\epsilon_\theta(\boldsymbol{x}^k, k)}{\sigma_k}$.

Wang et al. (2022); Chi et al. (2023) extend diffusion models as expressive and powerful policies for offline RL and robotics. In robotics, a set of past observation images $\mathbf{O}$ is used as input to the policy. An action chunk $\mathbf{A}$, which consists of a sequence of consecutive actions, forms the output of the policy. ResNet (He et al., 2016) based vision encoders are commonly utilized to encode multiple camera observation images into observation features. Diffusion policy is represented as a conditional diffusion-based action prediction model,

$$\pi_\theta(\mathbf{A}_t^0|\mathbf{O}_t) := \int \cdots \int \mathcal{N}(\mathbf{A}_t^K; \boldsymbol{0}, \boldsymbol{I}) \prod_{k=K}^{k=1} p_\theta(\mathbf{A}_t^{k-1}|\mathbf{A}_t^k, \mathbf{O}_t) d\mathbf{A}_t^K \cdots d\mathbf{A}_t^1,$$

where $\mathbf{O}_t$ contains the current and a few previous vision observation features at timestep $t$, and $p_\theta$ could be represented by $\epsilon_\theta$ as shown in DDPM (Ho et al., 2020). The explicit form of $\pi_\theta(\mathbf{A}_t^0|\mathbf{O}_t)$ is often impractical due to the complexity of integrating actions from $\mathbf{A}_t^K$ to $\mathbf{A}_t^1$. However, we can obtain an action chunk prediction $\mathbf{A}_t^0$ by iteratively solving Equation (9) from $K$ to 0.

Figure 11: Dynamic Real-World Experiment: Pose Reset.

# E    DISCUSSION

**Comparison with VSD.**    VSD is designed to distill image-level knowledge from powerful 2D priors, specifically pretrained text-to-image diffusion models, to facilitate 3D content generation. Its overarching objective—reverse KL optimization—is widely applied across multiple domains, including VAEs. In this work, we also apply reverse KL optimization for diffusion policy distillation. However, the implementation and derivation for different domains required major efforts. This extensive process involved adjustments to noise scheduling (DDPM and EDM), proper initialization, balancing the convergence of the generator and its score network, tuning parameters, designing experiments in dynamic environments, and conducting both simulated and real-world robotics experiments—an undertaking that should not be underestimated. Furthermore, OneDP considered temporal control characteristics by predicting action chunks, each comprising a sequence of actions (K=16). This approach addresses the temporal dependencies inherent in many robotics tasks, which are not considered in VSD.

**Training Cost Comparison of OneDP-D and OneDP-S.**    OneDP-S and OneDP-D differ in their computational requirements. The training cost for OneDP-S is approximately twice that of OneDP-D, due to the inclusion of the generator score network. When accounting for evaluation during training, the total time for OneDP-S is about 1.5 times longer than that of OneDP-D. For example, on the small dataset PushT, training and evaluation for OneDP-D take about 30 minutes, while OneDP-S requires approximately 45 minutes. On the larger ToolHang dataset, OneDP-D takes roughly 6 hours, compared to about 8 hours for OneDP-S. These details will be further elaborated in future revisions to provide a comprehensive view of the trade-offs between stochastic and deterministic policies in terms of both performance and computational efficiency.

# F    MORE DYNAMIC EXPERIMENTS

We conducted an additional dynamic real-world experiment to evaluate performance under human intervention. During the milk box pick-and-place task, we randomly reset the milk box pose to simulate changes in the environment. The process is illustrated in Figure 11. The results indicate that DP achieves a success rate of 28.57% (6/21), while our OneDP significantly outperforms it with a success rate of 76.19% (16/21), over 21 random initializations. DP fails in most cases due to its slow response to environmental changes, whereas OneDP reacts quickly and achieves a much higher success rate.

