# OpenReview forum: "One-Step Diffusion Policy: Fast Visuomotor Policies via Diffusion Distillation"
_ICLR.cc/2025/Conference — Submitted to ICLR 2025_

### Official Review · Reviewer_htHT · 2024-10-28

**Soundness:** 3
**Presentation:** 3
**Contribution:** 2
**Rating:** 6
**Confidence:** 3

**Summary:**

The paper proposed One-Step Diffusion Policy (OneDP) to reduce inference time through diffusion distillation. The method outperforms diffusion policy in both simulated and real-world environments.

**Strengths:**

- The proposed approach is straightforward and easy to follow.
- The method is evaluated on a wide range of tasks in both simulation and real-world.
- The paper is well written and easy to follow.

**Weaknesses:**

- The method of distillation follows the methodologies in image generation. The novelty is limited.
- Only one baseline (DDIM in Tab. 1 and CP in Tab. 2) is compared. There are many methods speeding up diffusion models that can be compared.

**Questions:**

- What are the limitations of OneDP?  (like ability on modeling mutimodality, capacity, etc.)
- Can the authors elaborates on why stochastic policies perform better in complex environments compared to deterministic ones? The stochastic policies seem to outperform deterministic ones in all the tasks.

---

> ### Author Response · Authors · 2024-11-15
> **Response to Reviewer htHT**
>
> We thank Reviewer htHT for the positive feedback and valuable suggestions. We provide further clarifications below.
>
> **W1:** The overarching objective, reverse KL optimization, is commonly applied across many domains, including text-to-3D and image generation. However, the implementation and derivation for different domains required major efforts. This extensive process involved adjustments to noise scheduling (DDPM and EDM), proper initialization, balancing the convergence of the generator and its score network, tuning parameters, designing experiments in dynamic environments, and conducting both simulated and real-world robotics experiments—an undertaking that should not be underestimated. Furthermore, OneDP demonstrates significant improvements in accelerating inference speed, achieving strong empirical results in both simulated environments and real-world experiments.
>
> **W2:** At the time of submission, CP was the most recent diffusion distillation work in robotics, employing consistency trajectory models. CP directly compares itself with DDPM and DDIM, so we followed the literature in comparing with DDPM, DDIM, and CP. Exploring additional methods to accelerate diffusion models for robotics applications is a promising direction for future work.
>
> **Q1:** We discuss OneDP’s limitations in Section 5. Currently, OneDP employs reverse KL for optimization, which may risk mode collapse; however, currently we found it performs well in most cases. Developing advanced distillation techniques remains a potential future direction. Additionally, our current diffusion policy operates at a moderate scale, so performance on large-scale and long-horizon tasks requires further experimental validation.
>
> **Q2:** In complex environments, the data distribution can be intricate and exhibit multimodal characteristics. For instance, in the coffee machine manipulation task, we intentionally collected data by teleoperating the arm to close the lid from either the left or right side, chosen randomly. In such cases, a stochastic policy better captures human-like behavior and handles uncertainties during evaluation. Theoretically, stochastic policies encompass determinstics policies (distribution estimate vs a point estimate). That Stochastic policies can outperform deterministic policies is also supported by numerous RL studies, such as SAC [1].
>
> [1] Haarnoja, Tuomas, Aurick Zhou, Pieter Abbeel, and Sergey Levine. "Soft actor-critic: Off-policy maximum entropy deep reinforcement learning with a stochastic actor." In *International Conference on Machine Learning*, pp. 1861-1870. PMLR, 2018.

---

> ### Author Response · Authors · 2024-11-20
> **Request for Feedback**
>
> Dear Reviewer htHT,
>
> Thank you for taking the time to review our work and provide your valuable feedback. We hope our responses have adequately addressed your concerns. If you have any further questions or wish to discuss any aspects of the paper, we would be glad to engage during the remaining discussion period.
>
> On the other hand, if you feel that your concerns have been fully resolved, we kindly request that you consider reevaluating your rating. Your support means a great deal to us and would contribute significantly to the recognition of our work.
>
> Thank you!

---

> > ### Comment · Reviewer_htHT · 2024-11-21
> >
> > Thank you for the detailed reply! While the rebuttal addresses most of my concerns, I find the methodology to be quite similar to those used in image diffusion models. I tend to maintain my score as a weak accept on this point. However, I am open to raising it if OneDP demonstrates the ability to solve non-trivial tasks that the original DP and CP methods could not address.

---

> ### Author Response · Authors · 2024-11-23
> **Response to Reviewer htHT**
>
> We thank Reviewer htHT for the additional feedback. In response, we have conducted a new dynamic experiment involving resetting the object pose during task execution. This task presents a significant challenge for DP, while OneDP demonstrates a consistently strong performance. The results show that **DP** achieves a success rate of **28.57% (20 trails)**, whereas our **OneDP** achieves a substantially higher success rate of **76.19% (20 trails)**. To provide further clarity, we have included a demo video of the pose-reset dynamic task, accessible at the following link: [https://drive.google.com/file/d/1AfaMXboHfblAME6SuCqaX8uj-SMiGDlA/view?usp=sharing](https://drive.google.com/file/d/1AfaMXboHfblAME6SuCqaX8uj-SMiGDlA/view?usp=sharing). We hope this addresses your concerns and kindly request that you consider reevaluating your rating based on the additional empirical evidence. These new results have been incorporated into our revision in Appendix F.

---

> > ### Comment · Reviewer_htHT · 2024-11-23
> >
> > The results look good! However, I don't find this task involves high dynamics that are fundamentally unsolvable for DP. I also wonder how well CP performs on this task. I would expect tasks requiring fast movement speeds or rapid feedback for contact-rich manipulation to pose more fundamental challenges for DP or even CP.

---

> > > ### Author Response · Authors · 2024-11-23
> > > **Response to Reviewer htHT**
> > >
> > > We thank the reviewer for accepting the good empirical results. We address your questions below.
> > >
> > > First, we believe that a success rate below 30% clearly demonstrates that DP is not suitable for this type of task. Second, this experiment involves dynamics requiring the robot to track and adapt to changes in the object pose. Given that DP struggles even with this level of dynamics, it is reasonable to expect that it would fail on tasks requiring greater dynamic adaptation.
> > >
> > > Designing dynamic experiments is indeed challenging. Notably, the CP paper does not include specific tasks focused on dynamics, and we have yet to see consistently high-dynamic tasks in robot manipulation research. We believe paper evaluations should be based on the current state of research. Our work introduces some dynamic tasks, which we believe represent a valuable point for advancing this area.
> > >
> > > For CP, CP’s public code does not provide setup details for real-robot execution. Our real-robot experiments were conducted using a different codebase, Robomimic. Implementing CP in Robomimic would require substantial additional effort. Since our primary goal is to evaluate our method, OneDP, and we have already compared it to CP in simulation tasks with clear outperforming, we leave the real-world comparison with CP for future work.

---

### Official Review · Reviewer_Xwvc · 2024-10-30

**Soundness:** 3
**Presentation:** 3
**Contribution:** 3
**Rating:** 6
**Confidence:** 3

**Summary:**

This work proposes an inference-efficient diffusion policy by distilling a pre-trained DP into an action generator with one-step inference. It greatly improves the practicality of DP methods in real-world tasks, especially for responsive tasks.

**Strengths:**

1. This work tackles a popular yet significant problem concerning accelerating inference in diffusion policy.
2. The authors conduct extensive experiments on visuomotor tasks in both simulation and real world.
3. The idea is straight-forward and the paper is well-written and easy to follow.

**Weaknesses:**

1. Consistency Policy (CP) [1] in the real world is missing in the experiment part, which can be a representative baseline considering accelerating DP. An additional experiment can be added to compare OneDP with CP in terms of inference speed and success rate.
2. More experiments on responsive or dynamic tasks can be added in the experiment part since potential application on responsive or dynamic tasks is a contribution of this work. A simple example can be randomly resetting the object pose sometime during the task execution.

[1] Prasad A, Lin K, Wu J, et al. Consistency policy: Accelerated visuomotor policies via consistency distillation[J]. arXiv preprint arXiv:2405.07503, 2024.

**Questions:**

1. A reverse KL is employed to align the distributions of the action generator and pre-trained DP, which is known to induce mode-collapse problems. How does the reverse KL affect the multi-modal property of DP?
2. I am confused about the fourth bar chart in Figure 1. Since OneDP requires first pre-training a DP, it seems confusing that the training cost of OneDP can be lower than DP itself; in other words, only considering the distillation procedure in OneDP can be an unfair comparison with DP in terms of training cost. Can you explain how you are measuring and comparing the training costs between OneDP and DP?

---

> ### Author Response · Authors · 2024-11-15
> **Response to Reviewer Xwvc**
>
> We thank Reviewer Xwvc for the positive feedback and valuable suggestions. Further clarifications are provided below.
>
> **W1:** We intended to include a comparison with Consistency Policy (CP) in real-robot experiments, but CP’s public code does not provide setup details for real-robot execution. Our real-robot experiments were conducted using a different codebase, Robomimic. Implementing CP in Robomimic would require substantial additional effort. Since our primary goal is to evaluate our method, OneDP, and we have already compared it to CP in simulation tasks, we leave the real-world comparison for future work.
>
> **W2:** We designed the ‘pnp-milk-move’ task to demonstrate responsive and dynamic interactions. We also provide **a video of coffee machine manipulation in dynamic environments** in the link in the abstract, where OneDP responds quickly and succeeds, while DP responds slowly and fails. We also tried other dynamic tasks, but most proved challenging to benchmark and standardize due to the human intervention cannot be guaranteed exactly the same everytime. The proposed ‘pnp-milk-move’ task, with its fixed trajectory, allows for a standardized process. We thank reviewer Xwvc for the suggestion. We are considering adding more dynamic experiments, and if the time permits, we will post the results here.
>
> **Q1:** Based on success rates observed in our experiments, reverse KL performs effectively in our current settings. For the coffee machine manipulation task, we deliberately collected data by teleoperating the arm to close the lid from either the left or right side, chosen at random. During evaluation, the distilled policy, OneDP, successfully captured this bimodal behavior, closing the lid from both sides. We also observe that reverse KL performs well in image generation [1-3].
>
> **Q2:** Yes, OneDP requires pre-training of DP. The columns in the fourth panel reflect only the distillation cost relative to pre-training cost. Our intent here is to emphasize that distillation “requires only a small fraction of the pre-training cost,” as stated in the caption. Thank you for noting this, and we will clarify this point in our revision.
>
> Reference:
>
> [1] Yin, Tianwei, et al. "One-step diffusion with distribution matching distillation." *Proceedings of the IEEE/CVF Conference on Computer Vision and Pattern Recognition*. 2024.
>
> [2] Nguyen, Thuan Hoang, and Anh Tran. "Swiftbrush: One-step text-to-image diffusion model with variational score distillation." *Proceedings of the IEEE/CVF Conference on Computer Vision and Pattern Recognition*. 2024.
>
> [3] Dao, Trung, et al. "SwiftBrush v2: Make Your One-step Diffusion Model Better Than Its Teacher." *European Conference on Computer Vision*. Springer, Cham, 2025.

---

> ### Author Response · Authors · 2024-11-20
> **Request for Feedback**
>
> Dear Reviewer Xwvc,
>
> Thank you for taking the time to review our work and provide your valuable feedback. We hope our responses have adequately addressed your concerns. If you have any further questions or wish to discuss any aspects of the paper, we would be glad to engage during the remaining discussion period.
>
> On the other hand, if you feel that your concerns have been fully resolved, we kindly request that you consider reevaluating your rating. Your support means a great deal to us and would contribute significantly to the recognition of our work.
>
> Thank you!

---

> > ### Comment · Reviewer_Xwvc · 2024-11-21
> >
> > Thanks for your reply! While many of my concerns have been addressed, I hold the opinion that more experiments on dynamic tasks should be included to demonstrate the power of OneDP to deal with responsive tasks. So I will keep my score, and I am willing to raise my rating if more empirical results can be posted.

---

> ### Author Response · Authors · 2024-11-23
> **Response to Reviewer Xwvc**
>
> We thank you for the additional feedback. In response to your suggestion, we have included an additional dynamic experiment involving resetting the object pose during task execution. The results demonstrate that the success rate for **DP** is **28.57% (20 trails)**, while our **OneDP** achieves a significantly higher success rate of **76.19% (20 trails)**. To illustrate the pose-reset dynamic task, we have provided a demo video, which can be accessed via the following link: https://drive.google.com/file/d/1AfaMXboHfblAME6SuCqaX8uj-SMiGDlA/view?usp=sharing. We hope this addresses your concerns and kindly request that you consider reevaluating your rating if you find the empirical results satisfactory. We have included the new empirical results in our revision in Appendix F.

---

### Official Review · Reviewer_XK7c · 2024-10-31

**Soundness:** 3
**Presentation:** 2
**Contribution:** 3
**Rating:** 6
**Confidence:** 3

**Summary:**

Summary:

This paper proposes a One-step Diffusion Policy (OneDP) that distills a pretrained diffusion policy into a single-step action generator by minimizing the KL divergence along the reverse chain.
Experiments on the robomimic benchmark and real-world tasks show that OneDP reduces inference time while maintaining state-of-the-art success rates.

**Strengths:**

Strengths


1) High inference time is a major limitation of diffusion policies. The proposed OneDP demonstrates less inference time through distillation.

2) OneDP introduces a weighted score estimation method from Diffusion-GAN to address the intractable KL divergence during the diffusion process.

3) The authors integrate the proposed distillation algorithm with two scheduling methods for discrete and continuous denoising time spaces.

**Weaknesses:**

Weaknesses

1) The contribution of this method is unclear, as score distillation sampling is well-explored in prior text-to-3D research. The paper does not discuss the differences between VSD and OneDP, nor does it consider the temporal control characteristic in OneDP.

2) Training an auxiliary diffusion network could increase computational costs and optimization complexity. Could the author provide further details on the computational costs and optimization complexity introduced by the generator score network?

3) Distilling to a deterministic policy may diminish the stochastic advantages of diffusion policies.

4) The claim that the student policy slightly outperforms the teacher policy is interesting. However, it requires further discussion and empirical validation.

**Questions:**

Questions

1) The clarity of the Method section could be enhanced by using consistent terminology for the auxiliary diffusion network in Lines 204 and 222. Additionally, consider removing some unnecessary subscripts in equations such as (5) and (8). Relocate expressions like $A_{\theta} = G_{\theta}(z, O), \quad z \sim N(0, I)$ to a "where..." explanation following the equation if needed.

2) About  “OneDP-S even slightly outperforms the pre-trained DP”.

    2.1 In Lines 345-347, the authors claim that “OneDP-S even slightly outperforms the pre-trained DP” and provide an explanation. However, this slight improvement may be due to stochastic effects caused by limited training seeds, as indicated by results in Square-mh, Square-ph, and Transport-ph in Table 1. Will this empirical ranking hold if the environment initialization state is expanded (both in the numbers and range)?

    2.2 Additionally, the authors' explanation requires further discussion. If the improvement stems from the pre-trained diffusion model (teacher) accumulating more errors, this suggests the teacher is not overfitted to the training set. Then, how could the student policy outperform the teacher? Including results on the training scenarios may offer further insights.

3) There is a lack of empirical results on inference efficiency. The authors should include comprehensive comparisons of inference efficiency between OneDP, the original diffusion model (DDPM), and other distillation methods like DDIM, CP, and Progressive Distillation (Salimans & H, 2022), using simulation experiments (if real-world ones are expensive).

---

> ### Author Response · Authors · 2024-11-15
> **Response to Reviewer XK7c**
>
> We thank Reviewer XK7c for the positive feedback and valuable suggestions. We provide further clarifications below.
>
> **W1:** VSD is developed to distill image-level knowledge from powerful 2D priors, specifically pretrained text-to-image diffusion models, to aid in 3D content generation. The overarching objective, reverse KL optimization, is commonly applied across many domains, including VAEs. However, the implementation and derivation for different domains required major efforts. This extensive process involved adjustments to noise scheduling (DDPM and EDM), proper initialization, balancing the convergence of the generator and its score network, tuning parameters, designing experiments in dynamic environments, and conducting both simulated and real-world robotics experiments—an undertaking that should not be underestimated. We have provided the detailed discussion regarding VSD in Appendix E. Furthermore, OneDP considered temporal control characteristics by predicting action chunks, each comprising a sequence of actions (K=16).
>
> **W2:** First, our distillation typically converges very quickly, within 20–100 epochs, resulting in a relatively short total training time. Second, in comparing OneDP-S (stochastic) with OneDP-D (deterministic, no additional score network), the training cost for OneDP-S is about twice as large. When including evaluation during training, the total training time for OneDP-S is approximately 1.5 times slower. For instance, on the small dataset PushT, OneDP-D takes around 30 minutes, while OneDP-S takes about 45 minutes, including all training and evaluation. On the larger ToolHang dataset, OneDP-D takes roughly 6 hours, and OneDP-S takes about 8 hours. We have provided the detailed discussion regarding computational cost in Appendix E.
>
> **W3:** While we agree with Reviewer XK7c’s point here, we emphasize we also provided stochastic one-step diffusion policy (OneDP-S). Through comparative experiments, we found that OneDP-S consistently outperformed OneDP-D, aligning with Reviewer XK7c’s hypothesis.
>
> **W4:** We also find the results intriguing and have provided explanations and empirical validations. Below, we further elaborate on these in our responses to the reviewer’s questions.
>
> **Q1:** We thank Reviewer XK7c for the constructive feedback, and we will revise the Method section to improve clarity.
>
> **Q2:** First, we observed that robotic task evaluations have high variance, so we increased the number of seeds used per task to 500, which is significantly more than the number of seeds used in Consistency Policy (N=200) and Diffusion Policy (N=150). OneDP consistently outperforms DP by a small margin across 5 out of 6 simulated tasks in the DDPM setting, as reflected in the average score. We, therefore, report a slight improvement. Second, we do not think this performance gain is related to overfitting. Iterative sampling methods can inherently suffer from accumulative errors in inference. As noted in the EDM2 [1] paper, “The final image quality is dictated by faint image details predicted throughout the sampling chain, and small mistakes at intermediate steps can have snowball effects in subsequent iterations.” However, applying the pretrained model as teacher network to guide and distill one-step student network avoids this issue, as the optimization occurs independently at each diffusion step, thereby circumventing the cumulative errors in iterative sampling.
>
> **Q3:** We clarify that the detailed inference efficiency comparison for real-world experiments is reported in Table 5, under our specific real-world compute setup.
>
> [1] Karras, Tero, Miika Aittala, Jaakko Lehtinen, Janne Hellsten, Timo Aila, and Samuli Laine. "Analyzing and improving the training dynamics of diffusion models." In *Proceedings of the IEEE/CVF Conference on Computer Vision and Pattern Recognition*, pp. 24174-24184. 2024.

---

> > ### Comment · Reviewer_XK7c · 2024-11-17
> >
> > Thank you for your detailed response and for addressing most of my concerns. The paper focuses on improving the inference speed of diffusion policy, which is a highly targeted, application-specific issue. I admire and appreciate the efforts the authors have made, particularly regarding real-world experiments. However, I noticed that the core idea still shares some similarities with VSD in the vision field, suggesting that the main contribution lies in the engineering adaptation from the vision domain to the robotics domain. Therefore, I may not increase the score further.

---

> ### Author Response · Authors · 2024-11-17
> **Response to Reviewer XK7c**
>
> First, we sincerely thank the reviewer for acknowledging our contributions and efforts, particularly in conducting experiments on real robots. **We would like to highlight that our primary submission area for ICLR is "Applications to Robotics, Autonomy, and Planning."** The ultimate goal of this work is to achieve fast and effective diffusion policies for real-world robotic systems, with a strong emphasis on demonstrating practical impact and advancing the field through real-world applicability.
>
> We respectfully argue that our paper should be evaluated not only for the novelty of its underlying machine learning methods, but also for how these methods are tailored to address real-world challenges. While our focus is on robotics systems, the challenges we tackle—such as delays between actions and outcomes—are likely generalizable to other real-world systems. Specifically, the application of reverse KL-based distillation to policy action spaces in robotics necessitated methodological innovations. This included deriving a distillation framework for action generation, addressing robotics-specific challenges such as balancing temporal control, managing multimodal data distributions, and ensuring real-time inference capabilities. These contributions go beyond simple engineering efforts and represent meaningful progress in adapting diffusion models to robotics.
>
> We believe our work offers a novel perspective on accelerating diffusion policies for robotics, laying the groundwork for further research in this domain and broader real-world applications.
>
> It is worth noting that just as **Diffusion Policy** [1] adapts diffusion models for reinforcement learning [2] in robotics, and **Consistency Policy** [3] applies Consistency Trajectory Models [4] to robotics, our work builds on foundational principles while addressing domain-specific challenges. This alignment with existing research demonstrates that our contribution not only fits within but also expands the body of work advancing robotics through innovative adaptations of diffusion models.
>
> We would sincerely appreciate it if the reviewer could reconsider the score, acknowledging that our work not only delivers practical advancements but also opens new research directions in the application of diffusion-based models to robotics and other systems with delayed response dynamics.
>
> [1] Chi, Cheng, et al. "Diffusion policy: Visuomotor policy learning via action diffusion." *The International Journal of Robotics Research* (2023): 02783649241273668.
>
> [2] Wang, Zhendong, Jonathan J. Hunt, and Mingyuan Zhou. "Diffusion policies as an expressive policy class for offline reinforcement learning." *arXiv preprint arXiv:2208.06193* (2022).
>
> [3] Prasad, Aaditya, et al. "Consistency policy: Accelerated visuomotor policies via consistency distillation." *arXiv preprint arXiv:2405.07503* (2024).
>
> [4] Kim, Dongjun, et al. "Consistency trajectory models: Learning probability flow ode trajectory of diffusion." *arXiv preprint arXiv:2310.02279* (2023).

---

> ### Author Response · Authors · 2024-11-20
> **Request for Feedback**
>
> Dear Reviewer XK7c,
>
> Thank you for taking the time to provide additional feedback. We kindly encourage you to review our latest responses. We believe our paper makes a meaningful and sound contribution to the "Applications to Robotics, Autonomy, and Planning" domain at ICLR. If you feel that your concerns have been fully resolved, we kindly request that you consider reevaluating your rating. Your support means a great deal to us and would contribute significantly to the recognition of our work.
>
> Thank you!

---

> > ### Comment · Reviewer_XK7c · 2024-11-25
> >
> > Thank you for your detailed response. After careful consideration and review of the comments from other reviewers, I have decided not to increase the score for your paper. This decision was not made lightly and is based on several key reasons:
> >
> > 1. Although this submission area is Applications to Robotics, Autonomy, and Planning, having real-world experiments does not automatically guarantee acceptance. I appreciate that you have validated the algorithm's effectiveness in practical applications, so I give the manuscript a positive score. But I find it challenging to further increase the score given the limited novelty of the paper.
> >
> > 2. Additionally, the impact of this method may be diminished by developments in other fields, such as more advanced diffusion methods (Score-based Diffusion Models) [1], more powerful hardware, and the progression of non-diffusion models like ACT [2].
> >
> > 3. The paper primarily focuses on small-scale, single-task model experiments. However, in practical applications, there is a greater need for inference acceleration in larger models that are used for pre-training or multitasking.
> >
> > 4. Another reviewer, htHT, also expressed concerns similar to mine regarding the core similarities between this method and image diffusion models.
> >
> > 5. Regarding the additional new dynamic real-world experiment, I found the results to be somewhat puzzling. When I multiply the total number of trials (20) by the success rates (28.57 and 76.91), the results are not integers. Could you please clarify how the success rates were calculated?
> >
> > I hope this feedback is helpful and constructive. Thank you for your understanding.
> >
> > [1] Reuss, Moritz, et al. "Goal-conditioned imitation learning using score-based diffusion policies." arXiv preprint arXiv:2304.02532 (2023).
> > [2] Zhao, Tony Z., et al. "Learning fine-grained bimanual manipulation with low-cost hardware." arXiv preprint arXiv:2304.13705 (2023).

---

> ### Author Response · Authors · 2024-11-25
> **Response to Reviewer XK7c**
>
> We thank Reviewer XK7c for the additional response. We would like to provide further clarifications below:
> 1. We think our paper aligns well with the "Applications to Robotics, Autonomy, and Planning" domain of ICLR. Similar to works like Diffusion Policy [1], Consistency Policy [2], and 3D Diffusion Policy [3], our method addresses the critical challenge of improving inference speed for diffusion models in robotics.
> 2. Diffusion models are inherently score-based generative models [4], and we see no fundamental obstacle to applying our method to accelerate score-based diffusion policies, such as those presented in [5]. Regarding Action Chunking with Transformers (ACT) [6], we acknowledge ACT as an excellent policy architecture that can complement diffusion-based models. For instance, DiT [7] demonstrates the innovative integration of transformers into diffusion models, achieving state-of-the-art results. Similarly, our method could be seamlessly extended to leverage such architectural advancements.
> 3. We also eagerly want to apply our method to multitask and large-scale diffusion policies. However, as of now, there is a lack of large-scale multitask diffusion-based foundation models in robotics. One key challenge of scaling up diffusion policies is their relatively slow action execution. We believe that our proposed approach, when combining pre-training and distillation, addresses this bottleneck and provides a promising pathway for future developments in this area.
> 4. We have addressed related concerns in our earlier responses, including the distinct contributions of our work and the methodological innovations tailored specifically for real-world robotic applications.
> 5. Thank you for pointing this out! We apologize for the misunderstanding. The actual evaluation consisted of 21 trials, with DP achieving a success rate of 28.57% (6/21) and OneDP achieving 76.19% (16/21). We mistakenly extrapolated the probabilities to 20 rounds, leading to a discrepancy. We have corrected this typo in the global response and the revised manuscript.
> We appreciate the reviewer’s thoughtful feedback and hope these clarifications provide a more comprehensive understanding of our contributions and their significance.
>
> [1] Chi, Cheng, et al. "Diffusion policy: Visuomotor policy learning via action diffusion." The International Journal of Robotics Research (2023): 02783649241273668.
>
> [2] Prasad, Aaditya, et al. "Consistency policy: Accelerated visuomotor policies via consistency distillation." arXiv preprint arXiv:2405.07503 (2024).
>
> [3] Ze, Yanjie, et al. “3D diffusion policy." arXiv preprint arXiv:2403.03954 (2024).
>
> [4] Song, Yang, et al. "Score-Based Generative Modeling through Stochastic Differential Equations." International Conference on Learning Representations.
>
> [5] Reuss, Moritz, et al. "Goal-conditioned imitation learning using score-based diffusion policies." arXiv preprint arXiv:2304.02532 (2023).
>
> [6] Zhao, Tony Z., et al. "Learning fine-grained bimanual manipulation with low-cost hardware." arXiv preprint arXiv:2304.13705 (2023).
>
> [7] Peebles, William, and Saining Xie. "Scalable diffusion models with transformers." Proceedings of the IEEE/CVF International Conference on Computer Vision. 2023.

---

### Official Review · Reviewer_HYTe · 2024-11-03

**Soundness:** 3
**Presentation:** 2
**Contribution:** 2
**Rating:** 5
**Confidence:** 3

**Summary:**

How can we distill diffusion policies into fast, one-step action generators? The authors propose One-Step Diffusion Policy (OneDP), a distillation-based approach to improve inference time of diffusion policies for robotics tasks from 1.5Hz to 62Hz. Diffusion policies have shown impressive results in imitation, and this work improves imitation learning by allowing for more responsive control due to fast inference. In order to distill Diffusion Policy, the authors opt for an SDS-based approach rather than one based on Consistency Models. The authors learn a generator (initialized from the pretrained imitation policy), which is updated via an SDS loss. For stochastic One-DP, an addition network is learned to approximate the score.

**Strengths:**

The paper is relatively straightforward. The contribution presented in this paper is limited, because the tasks presented don’t necessarily require such dynamic control and fast inference. However, the results seem reasonable, and the distillation approach is different from Consistency Policy.

OneDP consistently outperforms Consistency Policy. It matches DDPM / EDM, for the most part.

**Weaknesses:**

1. The contribution is limited. Although faster inference is a worthy contribution for imitation learning, none of the tasks, neither sim nor real, necessitate faster inference. If existing imitation learning algorithms can solve these tasks with their existing control frequency, what necessitates faster inference? What would be convincing is if there were tasks that genuinely required responsive, closed-loop control.
2, Figure 2 is unclear. In part a, it is unclear what the action generator is optimizing exactly. Part b seems to make more sense, although it is not immediately clear from the figure what the authors are trying to convey. The paper could benefit from a clearer diagram on both ends.
3, The input and output of the generator and action generator are unclear.
4. In the introduction, it is unclear the difference between the generator and action generator. Which one is actually being used as the policy?
5. Typo Line 212: “prertrained” should be “The parameters of the pretrained diffusion policy are fixed”

The authors should address the questions and weaknesses.

**Questions:**

1. In Section 2.2 Line 203, why can’t you obtain the score for Equation 1? Why do you have to learn an auxiliary diffusion network? This feels unnecessary and computationally burdensome, if you are training three separate identical diffusion models (since the generator and action generator are initialized from the pretrained DP).
2. What is the input and output of the generator? The generator is defined as an implicit policy and a one-step policy. Since the generator is initialized from Diffusion Policy parameters, is the time embedding (e.g. noise timestep 0, … 100) simply set always to 1? Or 100? Is the output just the predicted noise?
3. Why not learn an explicit policy and use the SDS loss?
4. How would distilling Diffusion Policy into a smaller DP model perform? For instance, reduce the number of parameters for DP and train it to match the pretrained DP outputs. DDIM with 10 steps should be faster due to the smaller size of the smaller DP model.
5. What is a deterministic action generator? Is z simply set to 0?
6. Is Diffusion Policy natively 1Hz? Even large VLAs like OpenVLA or Octo can run at 5Hz… Where exactly does this number come from?

---

> ### Author Response · Authors · 2024-11-15
> **Response to Reviewer HYTe**
>
> We thank Reviewer HYTe for providing detailed feedback. We believe there may be a misunderstanding regarding our paper. We first address the major concern in the contribution of our OneDP.
>
> **W1-1:** We respectfully but firmly disagree with this assessment. Fast inference is critical for robot manipulation tasks and visuomotor policies, as it enables smoother responses to dynamic visual inputs and resolves uncertainties in real-time during task execution. This capability directly impacts success rates and reduces task completion times, as evidenced by our results in Figure 1, Table 4, and Figure 7. Furthermore, our experiement set up follows the literature. Our experiments cover all major tasks from Consistency Policy, including both simulated and real-world tasks. In addition, we introduce the novel ‘pnp-milk-move’ task, specifically designed to test the benefits of faster inference for improved performance. This task is visualized in Figure 9, and Table 3 clearly demonstrates that faster inference leads to substantial performance gains. We also provide **a video of coffee machine manipulation in dynamic environments** in the link in the abstract, where OneDP responds quickly and succeeds, while DP responds slowly and fails.
>
> The remaining points are about the clarity of technical details, which we are happy to answer in details. Below, we address them point by point.
>
> **W1-2:** We will revise Figure 2 to improve clarity. In the current version, part (a) illustrates action prediction from our action generator (without optimization), while part (b) depicts the optimization of the full pipeline.
>
> **W1-3:** The inputs and outputs of our action generator are provided in Equation (3). There is no additional generator in our pipeline.
>
> **W2:** Our pipeline consists of three components: (1) an action generator, (2) a generator score network (for the action generator), and (3) a pretrained score network. Components (2) and (3) are only used during training and are discarded at inference. The action generator alone performs one-step inference for faster processing.
>
> **W3:** Thank you for noting this; we will correct the typo, ‘prertrained’ → ‘pretrained.’
>
> **Q1:** Our objective is to minimize the reverse KL divergence between the generator distribution and the pretrained model distribution for distillation. The gradient of the distillation loss takes the form of a score difference, necessitating score estimates from both distributions. The pretrained model, being a diffusion model, inherently provides these scores. Our one-step action generator is not a diffusion model, so we train an auxiliary score network to estimate its scores. Throughout training, only two networks, the action generator and the generator score network, are jointly optimized.
>
> **Q2:** The inputs include pure noise, a fixed initial timestep for DDPM, a fixed initial sigma value for EDM, and observations. The output for DDPM is the predicted noise, while for EDM, it’s the predicted action. The predicted noise can easily transform into an action prediction, so for consistency, we denote it as action prediction in Equation (3). In Table 7, we detail the hyperparameters for our initial time embedding, which are fixed and consistent across all experiments. For DDPM, we use $t_{init}=65$, while for EDM, we use $\sigma=2.5$.
>
> **Q3:** Explicit policies may struggle to capture complex data distributions, as Gaussian policies, for instance, are strictly symmetric and unimodal. Implicit generators, such as GANs, diffusion models, and flow-based models, have demonstrated advantages in diverse domains, including image generation, robotics, and RL. For a deterministic action generator, as shown in Equation (8), this loss resembles the SDS loss and does not require an additional generator score network. We compare OneDP-S and OneDP-D across all experiments and observe that stochastic policies generally perform better than determinsitic policies.
>
> **Q4:** We acknowledge that initialization, as noted in [1] and [2], is important for ensuring fast and stable convergence for distillation. Distilling into smaller models will break the initialization requisite. We agree that distilling into smaller models is a promising area and this singly remains an active research topic. We leave this aspect for future exploration.
>
> **Q5:** Yes, we set $ z = 0 $.
>
> **Q6:** This depends on the available computational resources. In our real-world experiment, we use a local desktop with one V100 GPU, with time costs detailed in Table 5. Frequency value comparisons are derived from Table 5.
>
> References:
>
> [1] Sauer, Axel, et al. "Adversarial diffusion distillation." European Conference on Computer Vision. Springer, Cham, 2025.
>
> [2] Xu, Yanwu, et al. "Ufogen: You forward once large scale text-to-image generation via diffusion gans." Proceedings of the IEEE/CVF Conference on Computer Vision and Pattern Recognition. 2024.

---

> ### Author Response · Authors · 2024-11-20
> **Request for Feedback**
>
> Dear Reviewer HYTe,
>
> Thank you for taking the time to review our work and provide your feedback. We hope our responses have adequately addressed your concerns. If you have any further questions or wish to discuss any aspects of the paper, we would be glad to engage during the remaining discussion period.
>
> On the other hand, if you feel that your concerns have been fully resolved, we kindly request that you consider reevaluating your rating. Your support means a great deal to us and would contribute significantly to the recognition of our work.
>
> Thank you!

---

> > ### Comment · Reviewer_HYTe · 2024-11-26
> > **Response to Authors**
> >
> > **Q1** In line 263/264 of your updated pdf, "We warm-start both the stochastic and deterministic action generator" -- this is why I assumed your action generator was trained with a diffusion objective. In any case, since predicting x_0 is equivalent to a transformation of predicting eps_0, couldn't there be a way to extract the scores?
> >
> > **Q2** Thank you for the explanation! I feel like it could be helpful in the paper to make clear what the input and output are, as you explained here. How did you choose t_init exactly? Was it just a random sweep, or is there a more principled reason why that value worked?
> >
> > **Q4** This makes sense, and I agree it is an orthogonal direction. It still would be interesting to see, but I don't think it is necessary as a baseline.
> >
> > Thank you to the authors for addressing my comments: it has been very helpful! At the moment, I plan on keeping my score.

---

> ### Author Response · Authors · 2024-11-26
> **Response to Reviewer HYTe**
>
> We thank Reviewer HYTe for engaging in the discussion. We are eager to address your concerns and questions. Below, we provide point-by-point responses:
>
> **Q1:** First, while we initialize the action generator from a diffusion model, it is not trained as a diffusion model; otherwise, it would not have the capability for one-step generation. Our action generator is a one-step model that takes noise as input and outputs clean actions.
> Second, it is correct that $x_0$ predictions can be derived from $\epsilon$ predictions through a transformation. However, in diffusion model theory, the $x_0$ prediction represents $\mathbb{E}[x_0 | x_t]$, which is not equivalent to the clean action prediction in our case. For instance, when $t$ is large (e.g., $t = 80$), $\mathbb{E}[x_0 | x_t]$ is a vague mean estimation of all possible $x_0$ values that could be reconstructed by reversing the diffusion chain from $x_t$. Therefore, clean action predictions $x_0$ cannot be directly extracted from the original diffusion model predictions without reversing the entire chain.
> Third, because our proposed action generator is purely implicit, we train an additional network to obtain the score estimates at any noise levels.
>
> **Q2:** We have revised the paper and clarified the inputs and outputs on page 5. We start from a large value for $t_{\text{init}}$ because higher $t_{\text{init}}$ values correspond to inputs closer to pure noise. We tested a small range of values, such as $[80, 65, 50]$, and fixed it at 65 as it already performed well.
>
> **Q4:** We thank you for your agreement on this point.
>
> We hope our response has addressed your concerns. Please feel free to reach out if you have any further questions or feedback.

---

> > ### Comment · Reviewer_HYTe · 2024-12-03
> > **Reponse to Authors**
> >
> > **Q1:** I agree with this point, but since the action generator is a one-step action generator (there are only noise steps $t=0$ and $t=1$, are scores still unable to be extracted?
> >
> > **Q2:** Thank you for the explanation.

---

> > > ### Author Response · Authors · 2024-12-03
> > > **Response to Reviewer HYTe**
> > >
> > > We thank Reviewer HYTe for agreeing with most of our explanations and appreciate the opportunity to clarify further regarding Q1.
> > >
> > > **Q1:** The one-step action generator is designed to predict clean actions given a fixed noise level $t=65$ and pure noise as the inputs. Once the clean action samples are predicted, they are perturbed across the full range of diffusion noise levels ($t=0, 1, \dots, 100$) to generate noisy predicted actions. To perform distillation, we need to estimate the scores for all $t$ values, not just $t=0$ or $t=1$.
> > >
> > > Moreover, even for a specific noise level such as $t=50$, extracting the score directly is challenging because the one-step action generator represents an implicit distribution after training. This implicit nature makes it difficult to directly compute the score without additional modeling.

---

> > > > ### Comment · Reviewer_HYTe · 2024-12-03
> > > > **Response to Authors**
> > > >
> > > > Thank you for the clarifications! I agree with Reviewer htHT and Xwvc that more dynamic tasks would be important to show the effectiveness of one-step diffusion policy. Thank you to the authors for their thoughtful responses and experiments during the rebuttal phase.

---

> > > > > ### Author Response · Authors · 2024-12-03
> > > > > **More dynamic tasks have been provided**
> > > > >
> > > > > Dear Reviewer HYTe,
> > > > >
> > > > > We are pleased to hear that our clarifications have been helpful. In response to the feedback received, we had already performed additional dynamic tasks that we believe are highly challenging for existing algorithms. If you have any additional specific dynamic tasks in mind that are within the current capabilities of state-of-the-art methods, we would greatly appreciate your suggestions.
> > > > >
> > > > > Thank you,
> > > > >
> > > > > Authors of OneDP

---

### Author Response · Authors · 2024-11-25
**New Dynamic Real-World Experiment**

Thank you to the reviewers for their valuable suggestions! In response, we have added a new and challenging dynamic real-world robot manipulation experiment. During the milk box pick-and-place task, we randomly reset the milk box pose to simulate changes in the environment. The results show that **DP** fails in most cases, achieving a success rate of only **28.57% (6/21)**, while our **OneDP** significantly outperforms it with a success rate of **76.19% (16/21)**, evaluated over 21 random initializations. DP struggles due to its slow response to environmental changes, whereas OneDP demonstrates quick adaptability, resulting in a much higher success rate. To provide additional context, we have included a demo video of the pose-reset dynamic task, available at the following link: https://drive.google.com/file/d/1AfaMXboHfblAME6SuCqaX8uj-SMiGDlA/view?usp=sharing. This new experiment has been detailed in Appendix F of our revised submission.

---

### Meta-Review · Area_Chair_rqQk · 2024-12-21

**Metareview:**

The paper proposes a One-Step Diffusion Policy (OneDP), a method that distills a pretrained diffusion policy (DP) into a fast, one-step action generator, significantly improving the inference time of diffusion policies. OneDP achieves inference speeds of 62Hz, making it suitable for real-world robotics tasks. The method uses score-based distillation and introduces an auxiliary diffusion network for training. The experimental results show that OneDP improves inference speed while maintaining competitive performance in various tasks, including both simulation and real-world settings.

Reasons to accept
- The paper is well-written and easy to follow.
- OneDP successfully accelerates diffusion policy inference, addressing a significant limitation of traditional diffusion models in real-time applications.
- The method is evaluated on a variety of tasks, both simulated and real-world, showcasing its practical potential.
- The method shows promising results, outperforming previous methods like Consistency Policy (CP) and DDIM in some cases.

Reasons to reject
- The distillation method is not particularly novel, as similar approaches have been used in other domains like image generation.
- The tasks evaluated do not convincingly require such fast inference speeds. The paper does not clearly explain why faster inference is necessary, especially when existing methods can already handle the tasks at lower frequencies.
- The experiments compare OneDP to only a few methods, and additional comparisons to other distillation techniques (e.g., Progressive Distillation) would strengthen the evaluation.
- The introduction of auxiliary diffusion networks and the need for additional networks in training raise questions about computational costs and optimization complexity, which are not fully addressed.
- (Minor) The revised paper violates the page limit (10 pages).

While this paper studies a promising research direction, i.e., accelerating the inference time of Diffusion Policy for controlling robots in real-world scenarios, and presents an interesting approach, the paper fails to convince the reviewers that the proposed method applies to high-dynamic tasks requiring higher inference speed. Consequently, I recommend rejecting the paper.

**Additional Comments On Reviewer Discussion:**

During the rebuttal period, all four reviewers acknowledged the author's rebuttal.

---

### Decision · Program_Chairs · 2025-01-22

Reject